# MITIGATING LABEL NOISE ON GRAPHS VIA TOPOLOGICAL CURRICULUM LEARNING

## ABSTRACT

Despite success on the carefully-annotated benchmarks, the effectiveness of graph neural networks (GNNs) can be considerably impaired in practice, as the real-world graph data might be noisily labeled. As a promising way to combat label noise, curriculum learning has gained significant attention due to its merit in reducing noise influence via a simple yet effective *easy-to-hard* training curriculum. Unfortunately, the early studies focus on i.i.d data, and when moving to non-iid graph data and GNNs, two notable challenges remain: (1) the inherent over-smoothing effect in GNNs usually induces the under-confident prediction, which exacerbates the discrimination difficulty between easy and hard samples; (2) there is no available measure that considers the graph characteristic to promote informative sample selection in curriculum learning. To address this dilemma, we propose a novel robust measure called *Class-conditional Betweenness Centrality* (CBC), designed to create a curriculum scheme resilient to graph label noise. The CBC incorporates topological information to alleviate the over-smoothing issue and enhance the identification of informative samples. On the basis of CBC, we construct a *Topological Curriculum Learning* (TCL) framework that guides the model learning towards clean distribution. We theoretically prove that TCL minimizes an upper bound of the expected risk under target clean distribution, and experimentally show the superiority of our method compared with state-of-the-art baselines.

## 1 INTRODUCTION

Noisy labels ubiquitous in real-world applications (Deng et al., 2020; Mirzasoleiman et al., 2020; Gao et al., 2022) inevitably impair the learning efficiency and the generalization robustness of deep neural networks (DNNs) (Rolnick et al., 2017; Nguyen et al., 2019). It becomes exacerbated on the graph data, as the noise influence can be propagated along the topological edges, unlike the independent and identically distributed (i.i.d.) data in the forms of image (Mirzasoleiman et al., 2020; Kim et al., 2021; Chen et al., 2019; Frénay and Verleysen, 2013; Thulasidasan et al., 2019; NT et al., 2019; Wei et al., 2021; Cheng et al.; Berthon et al., 2021). Combating the degeneration of GNNs on the noisily labeled graph then emerges as a non-negligible problem, drawing more attention from the research community (Dai et al., 2021; Li et al., 2021; Du et al., 2021; Yuan et al., 2023a;b; Xia et al., 2023).

Curriculum learning (Bengio et al., 2009) has been demonstrated as a promising way to deal with label noise on i.i.d. data (Han et al., 2018; Jiang et al., 2018; Zhou et al., 2020), owing to its simple yet effective *easy-to-hard* training curriculum. It builds upon the law of memorization effect that clean simple samples will be learned prior to clean hard samples and noise, which allows to design strategies of sample selection towards clean data regime (Arpit et al., 2017; Cheng et al., 2020; Northcutt et al., 2021), thereby mitigating the negative impact posed by corrupted labels. However, when applying curriculum learning on non-iid data, the *over-smoothing* (Chen et al., 2020; Keriven, 2022) characteristic of GNNs poses a new challenge in the construction of curriculum. This issue, characterized by increasing similarity in node features with growing network depth, results in a narrowed distinctions between easy and hard samples. Notably, this complication persists even in shallow GNN architectures, leading to under-confident predictions and more difficulty in the development of an easy-to-hard training curriculum, a concern echoed in previous studies (Wang et al., 2021a; Hsu et al., 2022). Besides, there is a scarcity of a robust measure that considers the topological characteristic within a noisily labeled graph to promote informative sample selection.

To address this dilemma, we propose a robust Class-conditional Betweenness Centrality (CBC) measure to efficiently distinguish the easy and hard nodes with topological enhancement for curriculum learning. As illustrated in Fig. 1, nodes located near topological class boundaries are much informative compared to nodes located far from topological class boundaries, as they may link nodes from diverse classes (Brandes, 2001; Barthelemy, 2004; Freeman, 1977; Zhu et al., 2020). However, those boundary-near nodes are hard to learn and identify in noisy labeled graphs due to the aggregation from the heterogeneous neighbours in GNNs (Bai and Liu, 2021; Wei et al., 2023). Building on this topological perspective, the proposed CBC evaluates the topological position of nodes by quantifying the heterogeneous message passing across different classes. This measurement is inspired by the *random-walk*

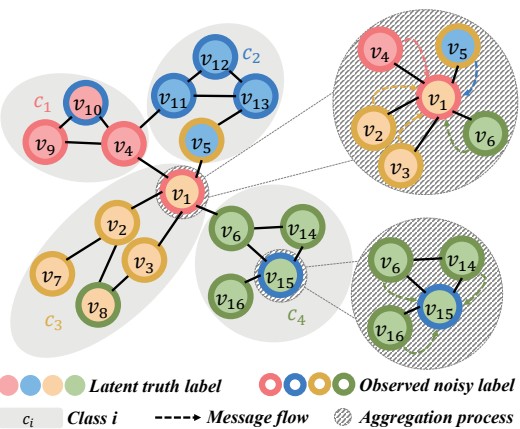

Figure 1: Illustration of node difficulty with noisy labels. $v_1$ is a hard node with corrupted label while $v_{15}$ is an easy node with corrupted label.

technique on graph structure (Nikolentzos and Vazirgiannis, 2020; Kang et al., 2012) rather than GNN predictions. Consequently, the CBC is impervious to the under-confident prediction issue caused by the over-smoothing effect in GNNs and showcases robustness in handling noise-induced confusion during predictions.

On the basis of CBC, we then propose a *Topological Curriculum Learning* (TCL) framework, which dynamically controls the selection of informative samples in the "easy-to-hard" curriculum to promote the training on noisily labeled graphs. Initially, the CBC measure drives the TCL to select clean and easy nodes located far from class boundaries. Despite providing less information, these nodes possess a high probability of being correctly labeled and easier to learn. Subsequently, we exploit clean nodes located in regions near class boundaries, guided by the CBC measure. These nodes in proximity to class boundaries with possible noisy labels, are more informative yet pose a learning challenge. Following this progressive curriculum, TCL emulates learning from an easier and cleaner pattern, gradually advancing to the harder yet more informative region. This framework smoothly guides the training of GNNs, mitigating the adverse effects of label noise, while simultaneously calibrating the model towards the clean and informative data regime (Wang et al., 2021b; Bengio et al., 2009). Our theoretical analysis further substantiates this claim. In a nutshell, our contributions can be summarized into the following points:

- We identify the challenge of the previous curriculum learning with the noisily labeled graph data, and propose a CBC measure to address the dilemma, which considers the topological characteristic to effectively and robustly select informative nodes for training under label noise.

- Based on the proposed CBC measure, we develop a novel TCL framework. This framework efficiently leverages the selected samples to progressively reduce the negative impact of label noise. We provide theoretical proof demonstrating that this framework consistently minimizes an upper bound of the expected risk under the target clean distribution.

- We conduct extensive experiments on various benchmarks to show the superiority of the proposed method over the state-of-the-art baselines in learning with noisily labeled graph, and provide comprehensive verification about the underlying mechanism of our method.

## 2 METHOD

### 2.1 NOTATIONS AND PRELIMINARY

Assume that we have an undirected graph $\mathcal{G} = (\mathcal{V}, \mathcal{E})$, where $\mathcal{V} = \{\mathbf{v}_1, ..., \mathbf{v}_n\}$ is the set of $n$ nodes, $\mathcal{E} \subseteq \mathcal{V} \times \mathcal{V}$ is the set of edges, and $\mathbf{A} \in \mathbb{R}^{n \times n}$ is the adjacency matrix of the graph $\mathcal{G}$. If nodes $\mathbf{v}_i$ and $\mathbf{v}_j$ are connected by edges $(\mathbf{v}_i, \mathbf{v}_j) \in \mathcal{E}$, $\mathbf{A}_{ij} = 1$; otherwise, $\mathbf{A}_{ij} = 0$. Let $\mathbf{D} \in \mathbb{R}^{n \times n}$ be the diagonal matrix, and $\hat{\mathbf{A}} \in \mathbb{R}^{n \times n}$ be the normalized adjacency matrix $\mathbf{D}^{-1/2}\mathbf{A}\mathbf{D}^{-1/2}$. Denote

$\mathcal{X} = \{\mathbf{x}_1, ..., \mathbf{x}_n\}$ and $\mathcal{Y} = \{y_1, ..., y_n\}$ as the sets of node attributes and node labels respectively, with $\mathbf{x}_i$ being the node attribute of node $\mathbf{v}_i$ and $y_i$ being the true label of node $\mathbf{v}_i$. In this study, we have a dataset $\mathcal{D} = \{\tilde{\mathcal{D}}_{\mathrm{tr}}, \mathcal{D}_{\mathrm{te}}\}$, where $\tilde{\mathcal{D}}_{\mathrm{tr}} = \{(\mathbf{A}, \mathbf{x}_i, \tilde{y}_i)\}_{i=1}^{n_{\mathrm{tr}}}$ is a noisy training set drawn from a noisy distribution $\mathbb{P}_{\tilde{\mathcal{D}}} = \mathbb{P}(\mathbf{A}, \mathcal{X}, \tilde{\mathcal{Y}})$ ($\tilde{y}_i$ is the noisy counterpart of $y_i$), and $\mathcal{D}_{\mathrm{te}}$ is a clean test set drawn from a clean distribution $\mathbb{P}_{\mathcal{D}} = \mathbb{P}(\mathbf{A}, \mathcal{X}, \mathcal{Y})$. Our goal is to learn a proper GNN classifier $f_{\mathcal{G}} : (\mathbf{A}, \mathcal{X}) \to \mathcal{Y}$ from the noisy training set $\tilde{\mathcal{D}}_{\mathrm{tr}}$.

## 2.2 CLASS-CONDITIONAL BETWEENNESS CENTRALITY

As discussed in the introduction, distinguishing between easy and hard samples on a noisily labeled graph is challenging due to the over-smoothing effect of GNNs. To combat this issue, we introduce a Class-conditional Betweenness Centrality (CBC) measure that takes into account the topological structure of nodes, formulated as follows.

**Definition 2.1** (Class-conditional Betweenness Centrality). *Given the Personalized PageRank matrix $\boldsymbol{\pi} = \alpha(\mathbf{I} - (1 - \alpha)\hat{\mathbf{A}})^{-1}$ ($\boldsymbol{\pi} \in \mathbb{R}^{n \times n}$), the Class-conditional Betweenness Centrality of the node $\mathbf{v}_i$ is defined by counting how often the node $\mathbf{v}_i$ is traversed by a random walk between pairs of other nodes that belong to different classes in a graph $\mathcal{G}$:*

$$\mathbf{Cb}_i := \frac{1}{n(n-1)} \sum_{\substack{\mathbf{v}_u \neq \mathbf{v}_i \neq \mathbf{v}_v, \\ \tilde{y}_u \neq \tilde{y}_v}} \frac{\boldsymbol{\pi}_{u,i} \boldsymbol{\pi}_{i,v}}{\boldsymbol{\pi}_{u,v}}, \tag{1}$$

*where $\boldsymbol{\pi}_{u,i}$ with the target node $\mathbf{v}_u$ and the source node $\mathbf{v}_i$ denotes the probability that an $\alpha$-discounted random walk from node $\mathbf{v}_u$ terminates at $\mathbf{v}_i$. Here an $\alpha$-discounted random walk represents a random traversal that, at each step, either terminates at the current node with probability $\alpha$, or moves to a random out-neighbour with probability $1 - \alpha$.*

Note that, CBC is inspired by the classical concept in graph theory – *Betweenness Centrality* (Newman, 2005; Brandes, 2001) that measures the centrality of nodes in a graph [1], but significantly differs from the class-conditional constraint and the random walk realization instead of the short-path counting. We kindly refer the readers to the Appendix A for the detailed discussion about their difference.

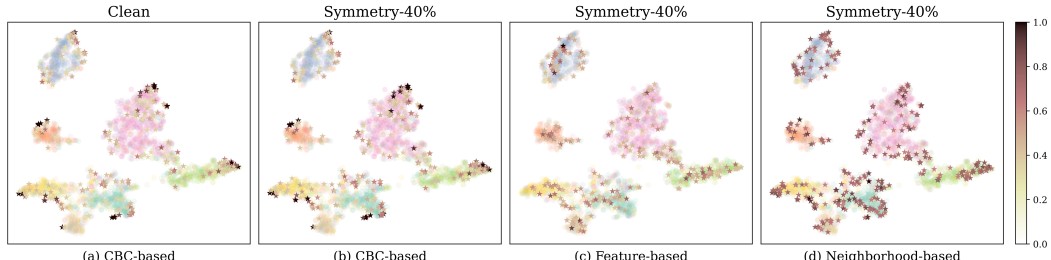

Figure 2: Robustness of Class-conditional Betweenness Centrality (*t-SNE* visualization of node embeddings based on trained GNNs from the *CORA* dataset). **(a)** clean labeled nodes with less CBC (lighter colour) are farther-away from class boundaries than those with high CBC (darker colour). **(b)(c)(d)** Compared with other two difficulty measurers (Wei et al., 2023; Li et al., 2023) in graph curriculum learning under 40% *Symmetric* label noise, CBC clearly shows superiority in terms of the differentiation *w.r.t.* boundary-near nodes.

**Robustness of Class-conditional Betweenness Centrality**  One promising merit of CBC is that it is robust to the label noise, although by definition it is based on the pair of nodes from different classes. As shown in Fig. 2 (b), under the high rate of label noise, the CBC of each node still can be accurately measured and the performance is close to the Fig. 2 (a) under clean labels. We also compare the performance of CBC with the other two difficulty measurers (Li et al., 2023) in the Figures 2 (c) and (d) to demonstrate our effectiveness. This is because CBC just requires that the node pairs belong to different classes instead of their absolutely accurate class labels, which is compatible

---

[1] In graph theory, the betweenness of a node $\mathbf{v}_i$ is defined to be the fraction of shortest paths between pairs of nodes in a graph that passes through $\mathbf{v}_i$. We provide its formal definition and discussion in Appendix.

with the general noise-agnostic scenarios. For example, if we have a pair of nodes whose latent true labels $(y_1 = 1, y_2 = 2)$ corresponding to the obvious noisy labels $(\tilde{y}_1 = 1, \tilde{y}_2 = 3)$, this node pair would not hinder the computation of CBC. Besides, even if the node pair actually belongs to the same underlying true class, CBC then degrades to the Betweenness Centrality and does not heavily hurt the total measure. Additionally, to demonstrate the consistent robustness of our CBC under varying levels of label noise, we visualize the superiority of our CBC distribution with numerical results as Fig. 3. The node dataset exhibits two distinct clusters, and despite a significant extent of label noise, certain nodes located near topological class boundaries consistently receive higher CBC scores. The complete and related experiment details have been presented in Appendix A.

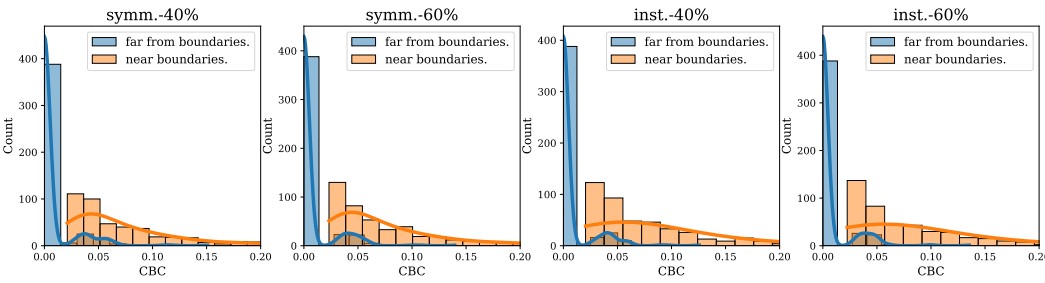

Figure 3: The distributions of the CBC score *w.r.t.* nodes on WikiCS with $40\%$ and $60\%$ symmetric noise (symm.) or $40\%$ and $60\%$ instance-based noise (inst.). The nodes are considered "far from topological class boundaries" (far from boundaries.) when their two-hop neighbours belong to the same class; conversely, nodes are categorized as "near topological class boundaries" (near boundaries.) when this condition does not hold. More comprehensive experiments can be seen in the Appendix A.

**Effectiveness of Class-conditional Betweenness Centrality** To demonstrate the effectiveness of Eq. (8), we conduct an empirical verification presented in Fig. 4. As observed, the ability to extract clean nodes from the subset of noisily labeled nodes notably diminishes as CBC increases, consistent with the expected behaviour of CBC. Additionally, nodes with elevated CBC values tend to be situated closer to the decision boundary, which is essential to characterize the decision boundary for classifier (Bengio et al., 2009; He et al., 2018; Huang et al., 2010; Vapnik, 1999; Bai and Liu, 2021). Leveraging the CBC measure allows us to selectively choose more informative nodes, significantly enhancing GNNs' performance during the training process. For further empirical evidence demonstrating the positive correlation between test accuracy and the overall CBC of the training set, we kindly refer readers to Appendix A.

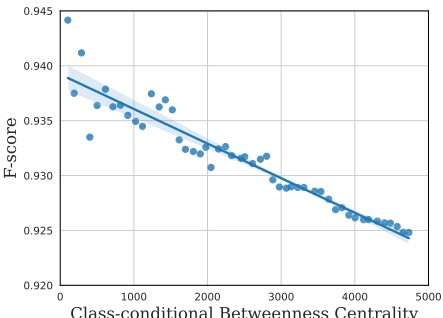

Figure 4: Correlation between *F-score* of extracting confident nodes and overall CBC of the noisily labeled subsets in a graph with $30\%$ *Symmetric* label noise. The Pearson coefficient is $-0.9276$ on 50 randomly selected subsets with $p$ value smaller than $0.0001$.

## 2.3 TOPOLOGICAL CURRICULUM LEARNING

In this section, we construct the Topological Curriculum Learning (TCL) framework leveraging the CBC measure to enhance training efficiency in the presence of label noise. Utilizing the CBC measure, the TCL process begins by training the model on easy clean nodes that are located far from class boundaries. Subsequently, the focus shifts to nodes closer to class boundaries that are hard and noisy labeled, with their identification guided by the CBC measure within the TCL framework. This process also could be further enhanced through integration with existing clean sample extraction techniques. It is important to note that the clean node extraction module within the TCL framework remains versatile and compatible with various clean sample selection technologies (Kim et al., 2021; Bai and Liu, 2021), and we will delve into the details shortly.

Here, we devise an "easy-to-hard" curriculum within our TCL framework, building upon the CBC. This curriculum is structured as a sequence of training criteria $\langle \tilde{Q}_\lambda \rangle$ with the increasing pace parameter $0 \le \lambda \le 1$. Each criterion $\tilde{Q}_\lambda$ is a reweighting of the noisy training distribution $\mathbb{P}_{\tilde{\mathcal{D}}}$. The early $\tilde{Q}_\lambda$

emphasises the easy nodes evaluated by CBC, and as $\lambda$ increases, more hard nodes are progressively added into $\tilde{Q}_\lambda$, detailed in the following. Note that, while several methods may involve in curriculum learning, few of them address noisy labeled graphs by considering the intricate graph structure [2].

**Extracting Clean Labeled Nodes** The extraction of clean labeled nodes is closely related to the *memorization effect* of neural networks (Arpit et al., 2017). Specifically, due to the memorization effect, the GNN classifier pre-trained at early epochs would fit the clean data well but not the incorrectly labeled data. We can treat the training nodes whose noisy labels are identical to the ones predicted by the pre-trained classifier as the confident nodes, indicating a higher likelihood of having clean labels (Bai and Liu, 2021). Note that, there are other similar rules to extract confident examples, e.g., those who have a high confidence score or corresponding to a smaller loss value (Han et al., 2018; Yu et al., 2019; Xia et al., 2021), which will be compared in experiments. Now, we progressively obtain the extracted confident node set $\mathbb{P}_{\hat{\mathcal{D}}}$ from $\mathbb{P}_{\tilde{\mathcal{D}}}$, which approximates nodes drawn from a target clean distribution $\mathbb{P}_{\mathcal{D}}$. With $\mathbb{P}_{\hat{\mathcal{D}}}$, we can construct our robust learning curriculum.

**Definition 2.2** (Topological Curriculum Learning). *Assume a sequence of confident training criteria $\langle \hat{Q}_\lambda \rangle$ with the increasing pace parameter $\lambda$. Each confident criterion $\hat{Q}_\lambda$ is a reweighting of the confident distribution $\mathbb{P}_{\hat{\mathcal{D}}}(z)$, where $z$ is a random variable representing an extracted confident node for the learner. Let $0 \leq W_\lambda(z) \leq 1$ be the weight on $z$ at step $\lambda$ in the curriculum sequence, and*

$$\hat{Q}_\lambda(z) \propto W_\lambda(z)\mathbb{P}_{\hat{\mathcal{D}}}(z), \tag{2}$$

*such that $\int_{\mathcal{Z}} \hat{Q}_\lambda(z)dz = 1$, where $\mathcal{Z}$ denotes the whole set of extracted confident nodes from each $\hat{Q}_\lambda(z)$. Then, the following two conditions are satisfied:*

- *(i) The entropy of distributions gradually increases, i.e., $H(\hat{Q}_\lambda)$ is monotonically increasing with respect to the increasing pace $\lambda$.*

- *(ii) The weight function $W_\lambda(z)$ for any confident nodes is monotonically increasing with respect to the increasing pace $\lambda$.*

In Definition 2.2, Condition (i) means that the diversity and the information of the confident set should gradually increase, *i.e*, the reweighting of nodes in later steps increases the probability of sampling informative nodes evaluated by CBC. Condition (ii) means that as gradually adding more confident nodes, the size of the confident node set progressively increases. Intuitively, in our TCL, the key is the proposed CBC measure that works as a difficulty measurer and defines the weight function $W_\lambda(z)$. This formalization has been widely used in the related curriculum learning literature (Bengio et al., 2009; Wang et al., 2021b). With the help of CBC, we can design a robust "easy-to-hard" learning curriculum that first extracts confident nodes from noisily easy nodes – that we term as *high-confident nodes* to train GNNs and then extracts confident nodes from noisily hard nodes – that we term as *low-confident nodes* to continually train. We summarize the procedure of TCL in Algorithm 1 of the Appendix and the utilization of CBC is akin to lines 3-6 of Algorithm 1.

## 2.4 THEORETICAL GUARANTEE OF TCL

Here, we first investigate the change in deviation between $\mathbb{P}_{\hat{\mathcal{D}}}$ and $\mathbb{P}_{\mathcal{D}}$ during the learning phases of TCL. Then, we theoretically prove that, with the $\mathbb{P}_{\hat{\mathcal{D}}}$, our TCL framework persistently minimizes an upper bound of the expected risk under target clean distribution.

Taking the binary node classification as an example, after extracting the confident nodes, our goal is to learn a proper GNN classifier $f_{\mathcal{G}} : (\mathbf{A}, \mathcal{X}) \rightarrow \mathcal{Y}$ with the input extracted confident nodes $z_i = \{(\mathbf{A}, \mathbf{x}_i, y_i)\}_{i=1}^{n_{cf}}$ from the confident distribution $\mathbb{P}_{\hat{\mathcal{D}}}(\mathcal{Z}) = \mathbb{P}_{\hat{\mathcal{D}}}(\mathbf{A}, \mathcal{X}|\mathcal{Y})\mathbb{P}_{\hat{\mathcal{D}}}(\mathcal{Y})$ (Cucker and Zhou, 2007), such that the following expected risk can be minimized:

$$\mathcal{R}(f_{\mathcal{G}}) := \int_Z \mathcal{L}_{f_{\mathcal{G}}}(z)\mathbb{P}_{\mathcal{D}}(\mathbf{A}, \mathbf{x}|y)\mathbb{P}_{\mathcal{D}}(y)dz, \tag{3}$$

where $\mathbb{P}_{\mathcal{D}}(\mathcal{Z}) = \mathbb{P}_{\mathcal{D}}(\mathbf{A}, \mathcal{X}|\mathcal{Y})\mathbb{P}_{\mathcal{D}}(\mathcal{Y})$ denotes the target clean distribution on $\mathcal{Z}$, and $\mathcal{L}_{f_{\mathcal{G}}}(z) = \mathbb{1}_{f_{\mathcal{G}}(\mathbf{A},\mathbf{x}) \neq y} = \frac{1 - yf_{\mathcal{G}}(\mathbf{A},\mathbf{x})}{2}$ denotes the loss function measuring the difference between the predictions and labels. Since the deduction for both $y = 1$ and $y = -1$ cases are exactly similar, we only consider

---

[2] More discussion of related works has been summarized in the Appendix B due to the space limitation.

one case in the following and denote $\mathbb{P}_{\mathcal{D}}(\mathbf{A}, \mathbf{x}) = \mathbb{P}_{\mathcal{D}}(\mathbf{A}, \mathbf{x}|y=1)$ and $\mathbb{P}_{\hat{\mathcal{D}}}(\mathbf{A}, \mathbf{x}) = \mathbb{P}_{\hat{\mathcal{D}}}(\mathbf{A}, \mathbf{x}|y=1)$. Let $0 \leq W_{\lambda^*}(\mathbf{A}, \mathbf{x}) \leq 1$, $\alpha^* = \int_{\mathbf{A}, \mathcal{X}} W_{\lambda^*}(\mathbf{A}, \mathbf{x}) \mathbb{P}_{\hat{\mathcal{D}}}(\mathbf{A}, \mathbf{x}) d\mathbf{x}$ denote the normalization factor[3] and $E(\mathbf{A}, \mathbf{x})$ measures the deviation from $\mathbb{P}_{\hat{\mathcal{D}}}(\mathbf{A}, \mathbf{x})$. Combining with Definition 2.2, we can construct the below curriculum sequence for theoretical evaluation (See proof in the Appendix C):

$$\hat{Q}_{\lambda}(\mathbf{A}, \mathbf{x}) \propto W_{\lambda}(\mathbf{A}, \mathbf{x}) \mathbb{P}_{\hat{\mathcal{D}}}(\mathbf{A}, \mathbf{x}), \tag{4}$$

where

$$W_{\lambda}(\mathbf{A}, \mathbf{x}) \propto \frac{\alpha_{\lambda} \mathbb{P}_{\mathcal{D}}(\mathbf{A}, \mathbf{x}) + (1-\alpha_{\lambda}) E(\mathbf{A}, \mathbf{x})}{\alpha^* \mathbb{P}_{\mathcal{D}}(\mathbf{A}, \mathbf{x}) + (1-\alpha^*) E(\mathbf{A}, \mathbf{x})}$$

with $0 \leq W_{\lambda}(\mathbf{A}, \mathbf{x}) \leq 1$ through normalizing its maximal value as 1 and $\alpha_{\lambda}$ varies from 1 to $\alpha^*$ with increasing pace parameter $\lambda$. Note that, the initial stage of TCL sets $W_{\lambda}(\mathbf{A}, \mathbf{x}) \propto \frac{\mathbb{P}_{\mathcal{D}}(\mathbf{A}, \mathbf{x})}{\mathbb{P}_{\hat{\mathcal{D}}}(\mathbf{A}, \mathbf{x})}$, which is of larger weights in the high-confident nodes while much smaller in low-confident nodes. With the pace $\lambda$ increasing, the large weights in high-confidence areas become smaller while small ones in low-confidence areas become larger, leading to more uniform weights with smaller variations.

Here, we introduce a *local-dependence* assumption for graph-structured data: Given the data related to the neighbours within a certain number of hops of a node $\mathbf{v}_i$, the data in the rest of the graph will be independent of $\mathbf{v}_i$ (Wu et al., 2020). This assumption aligns with Markov chain principles (Revuz, 2008), stating that the node is independent of the nodes that are not included in their two-hop neighbors when utilizing two-layer GNN, which does not means the totally i.i.d w.r.t. each node but means i.i.d w.r.t. subgroups. The local-dependence assumption is well-established and has been widely adopted in numerous graph theory studies (Schweinberger and Handcock, 2015; Didelez, 2008). It endows models with desirable properties which make them amenable to statistical inference (Schweinberger and Handcock, 2015). Therefore, based on the local-dependence assumption, for a node with the certain hops of neighbours $Z^{\mathbf{A}}$, after aggregation, we will obtain node representation $Z_{\mathbf{x}_i}$ that is approximately independent and identically distributed with nodes outside of $Z^{\mathbf{A}}$. We refer readers to (Gong et al., 2016) for more details. Finally, with Eq. (10) as the pace distribution, we have the following theorem and a detailed proof is provided in Appendix C.

**Theorem 1.** *Suppose $\{(Z_{\mathbf{x}_i}, y_i)\}_{i=1}^m$ are i.i.d. samples drawn from the pace distribution $Q_{\lambda}$ with radius $|X| \leq R$. Denote $m_+/m_-$ be the number of positive/negative samples and $m^* = \min\{m_-, m_+\}$. Let $\mathcal{H} = \{\mathbf{x} \to \mathbf{w}^T\mathbf{x} : \min_s|\mathbf{w}^T\mathbf{x}| = 1 \cap ||\mathbf{w}|| \leq B\}$, and $\phi(t) = (1-t)_+$ for $t \in \mathbb{R}$ be the hinge loss function. For any $\delta > 0$ and $g \in \mathcal{H}$, with confidence at least $1 - 2\delta$, have:*

$$\begin{aligned}
\mathcal{R}(sgn(g)) \leq &\frac{1}{2m_+} \sum_{i=1}^{m_+} \phi(y_i g(Z_{\mathbf{x}_i})) + \frac{1}{2m_-} \sum_{i=1}^{m_-} \phi(y_i g(Z_{\mathbf{x}_i})) \\
&+ \frac{RB}{\sqrt{m^*}} + 3\sqrt{\frac{\ln(1-\delta)}{m^*}} \\
&+ (1-\alpha_{\lambda})\sqrt{1 - \exp\{-D_{KL}(\mathbb{P}_{\mathcal{D}}^+||E^+)\}} \\
&+ (1-\alpha_{\lambda})\sqrt{1 - \exp\{-D_{KL}(\mathbb{P}_{\mathcal{D}}^-||E^-)\}},
\end{aligned} \tag{5}$$

*where $E^+$, $E^-$ denote error distributions that capture the deviation from $\mathbb{P}_{\mathcal{D}}^+$, $\mathbb{P}_{\mathcal{D}}^-$ to $\mathbb{P}_{\hat{\mathcal{D}}}^+$, $\mathbb{P}_{\hat{\mathcal{D}}}^-$.*

**Remark 1** (on the upper bound of the expected risk $\mathcal{R}(\text{sgn}(g))$. The error distribution $E$ reflects the difference between the noisy distribution and the clean distribution. Essentially, this error distribution serves as a bridge connecting the noisy and clean distributions in our upper bound. Thus, the last two rows measure the generalization capability of the learned classifier, which is monotonically increasing with respect to both the KL-divergence between the error distribution $E$ and the clean distribution $\mathbb{P}_{\mathcal{D}}$, and the pace parameter $\lambda$. That is, the less deviated is the error $E$ from $\mathbb{P}_{\mathcal{D}}$, the more beneficial is to learn a proper classifier from $\mathbb{P}_{\hat{\mathcal{D}}}$ which can generalize well on $\mathbb{P}_{\mathcal{D}}$.

Thus, the TCL process with curriculum $\hat{Q}_{\lambda}$ makes it feasible to approximately learn a graph model with minimal expected risk on $\mathbb{P}_{\mathcal{D}}$ through the empirical risk from $\mathbb{P}_{\hat{\mathcal{D}}}$, since the "easy-to-hard" property of the curriculum $\hat{Q}_{\lambda}$ intrinsically facilitates the information transfer from $\mathbb{P}_{\hat{\mathcal{D}}}$ to $\mathbb{P}_{\mathcal{D}}$. In

---

[3]The $\alpha^* \leq 1$ since $W_{\lambda^*}(\mathbf{A}, \mathbf{x}) \mathbb{P}_{\hat{D}}(\mathbf{A}, \mathbf{x}) \leq \mathbb{P}_{\hat{D}}(\mathbf{A}, \mathbf{x})$

specific, we can approach the task of minimizing the expected risk on $\mathbb{P}_{\mathcal{D}}$ by gradually increasing the pace $\lambda$, generating relatively high-confident nodes from $\hat{Q}_\lambda$, and minimizing the empirical risk on those nodes. This complies with the core idea of the proposed TCL. In addition, the first row in the upper bound of Theorem 1 corresponds to the empirical risk on training nodes generated from $Q_\lambda$. The second row reflects that the more training nodes are considered, the better approximation of expected risk can be achieved (Haussler and Warmuth, 1993; Haussler, 1990).

## 3 EXPERIMENTS

In this section, we conduct extensive experiments to verify the effectiveness of our method and provide comprehensive ablation studies about the underlying mechanism of our method.

**Datasets** We adopted three small datasets including *Cora*, *CiteSeer*, and *PubMed*, with the default dataset split as did in (Chen et al., 2018), and four large datasets including *WikiCS*, *Facebook*, *Physics* and *DBLP* to evaluate our method. Detailed statistics are summarized in Appendix. Following previous works (Dai et al., 2021; Du et al., 2021; Xia et al., 2020b), we consider three settings of simulated noisy labels, i.e, *Symmetric* noise, *Pairflip* noise and *Instance-dependent* noise. More explanation about these noise settings can be found in Appendix D.2.

**Baselines** We compare TCL with several state-of-the-art curriculum learning with noisy labels on i.i.d. data: (1) Co-teaching+ (Yu et al., 2019), (2) Me-Momentum (Bai and Liu, 2021) and (3) MentorNet (Yu et al., 2019). And we also compare the TCL with the graph curriculum learning method: (1) CLNode (Wei et al., 2023), (2) RCL (Zhang et al., 2023). Besides, some denoising methods on graph data have been considered (1) LPM (Xia et al., 2020a), (2) CP (Zhang et al., 2020), (3) NRGNN (Dai et al., 2021), (4) PI-GNN (Du et al., 2023), (5) RT-GNN (Qian et al., 2023) and (6)RS-GNN (Dai et al., 2022). More details about implementations are provided in the Appendix D.3.

### 3.1 MAIN RESULTS

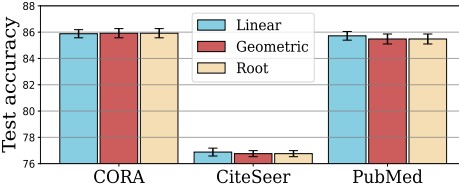 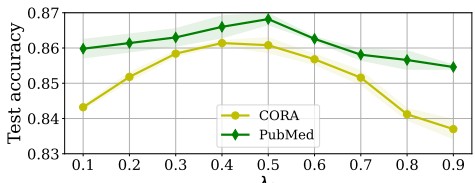

Figure 5: The hyperparameter analysis of TCL. The experiment results are reported over five trials under the 20% *Symmetric* noise. **(a)** The test accuracy of TCL with three different pacing functions on various datasets. **(b)** The test accuracy of TCL with increasing $\lambda_0$ on *CORA* and *PubMed*.

**Performance comparison on public graph datasets** Table 1 shows the experimental results on three synthetic noisy datasets under various types of noisy labels. For three datasets, as can be seen, our proposed method produces the best results in all cases. When the noise rate is high, the proposed method still achieves competitive results through the extraction of confident nodes. Although some baselines, *e.g.*, NRGNN, can work well in some cases, experimental results show that they cannot handle various noise types. In contrast, the proposed TCL achieves superior robustness against broad noise types. Some popular curriculum-learning-based methods that have worked well on learning with noisy labels on i.i.d. data, *e.g.*, Co-teaching+ and MentorNet do not show superior performance on graph data. This illustrates that the unique topology consideration in GNNs brings new challenges to those prior works and proves the necessity of our proposed method.

**Performance comparison on large graph datasets** We justify that the proposed methods can effectively alleviate the label noise on large graph datasets. Detailed descriptions of these graph datasets are provided in the Appendix. As shown in Table 2, our proposed method is consistently superior to other methods across all settings. Additionally, on certain datasets, labeled nodes are sparse *e.g.*, WikiCS that contains only $4.96\%$ labeled nodes or Physics that contains only $1.45\%$. The results indicate that our method is robust even in the presence of a small number of labeled nodes.

**Hyperparameter sensitivity** In TCL, the hyperparameter $\lambda$ affects the performance by controlling the construction of each curriculum. Correspondingly, the pacing function $\lambda(t)$ with training epoch

Table 1: Mean and standard deviations of classification accuracy (percentage) on synthetic noisy datasets with different noise levels. The results are reported over ten trials and the best are bolded.

| | Method | Symmetric | | | Pairflip | | | Instance-dependent | | |
|---|---|---|---|---|---|---|---|---|---|---|
| | | 30% | 40 % | 50% | 20% | 30% | 40% | 30% | 40% | 50% |
| CORA | Cross-Entropy | 83.61±1.07 | 80.86±1.46 | 75.14±2.44 | 82.23±0.93 | 75.87±1.20 | 62.05±3.59 | 83.21±0.74 | 80.32±0.94 | 74.96±1.82 |
| | LPM | 82.73±0.64 | 78.12±1.17 | 70.23±2.17 | 83.39±1.22 | 77.44±1.93 | 64.02±5.04 | 82.81±0.87 | 77.67±2.01 | 70.55±1.86 |
| | CP | 82.37±1.38 | 79.97±1.74 | 76.19±2.26 | 80.24±0.96 | 73.02±1.56 | 58.04±3.78 | 82.37±1.09 | 80.36±1.21 | 74.17±2.68 |
| | NRGNN | 81.73±1.80 | 79.08±3.18 | 77.36±2.03 | 81.83±0.93 | 77.10±1.52 | 64.13±3.98 | 81.62±2.08 | 78.66±2.54 | 76.31±2.98 |
| | PI-GNN | 82.48±0.10 | 80.36±0.10 | 77.59±0.20 | 83.10±0.10 | 77.96±0.20 | 63.62±0.30 | 81.83±1.00 | 80.02±1.07 | 77.27±1.21 |
| | RT-GNN | 83.21±1.05 | 80.46±1.06 | 75.84±1.43 | 82.53±0.73 | 76.87±1.09 | 61.75±2.29 | 82.14±0.97 | 80.13±1.23 | 74.82±0.94 |
| | RS-GNN | 83.21±0.29 | 79.00±0.15 | 77.21±0.43 | 81.83±0.37 | 76.46±0.24 | 63.09±0.22 | 82.83±0.73 | 78.93±0.63 | 76.52±0.83 |
| | Co-teaching+ | 82.59±0.96 | 79.81±1.30 | 74.59±2.33 | 81.70±1.45 | 75.59±2.13 | 59.03±5.76 | 81.84±1.10 | 79.70±1.34 | 73.36±2.54 |
| | Me-momentum | 83.76±0.25 | 81.82±0.72 | 79.48±0.63 | 84.09±0.48 | 78.04±1.03 | 64.07±1.03 | 83.14±0.25 | 82.04±0.57 | 77.33±0.82 |
| | MentorNet | 81.84±0.86 | 78.52±2.01 | 73.82±2.83 | 80.83±1.88 | 72.56±3.42 | 59.78±4.59 | 81.59±0.92 | 78.49±1.63 | 72.41±3.66 |
| | CLNode | 80.98±1.50 | 77.11±2.25 | 74.39±2.41 | 83.43±0.89 | 73.89±1.97 | 55.38±2.80 | 81.12±2.43 | 75.11±2.93 | 68.44±4.88 |
| | RCL | 73.20±0.12 | 76.36±1.09 | 63.40±0.73 | 71.06±0.48 | 65.30±0.80 | 51.34±0.42 | 69.20±1.00 | 59.30±0.13 | 54.16±2.25 |
| | **TCL** | **85.02±0.12** | **82.58±0.92** | **81.16±0.80** | **85.26±0.30** | **78.50±0.72** | **65.15±1.53** | **84.70±0.04** | **83.31±0.21** | **80.15±0.36** |
| CiteSeer | Cross-Entropy | 75.13±0.70 | 73.85±0.85 | 70.74±1.86 | 76.61±0.53 | 73.87±1.08 | 62.92±4.11 | 74.83±1.04 | 73.22±0.71 | 69.42±2.07 |
| | LPM | 73.19±1.07 | 69.54±1.37 | 61.22±2.08 | 75.08±0.76 | 69.91±1.31 | 56.56±6.50 | 73.55±0.79 | 69.32±1.76 | 61.90±1.73 |
| | CP | 73.26±1.22 | 70.99±1.88 | 63.74±2.55 | 74.36±1.21 | 68.21±2.56 | 56.56±6.50 | 73.45±0.72 | 69.90±1.64 | 64.61±2.74 |
| | NRGNN | 75.41±1.04 | 73.52±1.46 | 70.98±2.47 | 75.72±1.04 | 74.13±1.38 | 63.60±4.83 | 75.33±0.91 | 74.36±1.45 | 71.61±1.76 |
| | PI-GNN | 73.55±0.14 | 71.05±0.21 | 68.02±0.20 | 73.06±0.13 | 69.91±0.32 | 60.62±0.41 | 74.28±0.78 | 70.66±1.51 | 67.81±1.99 |
| | RT-GNN | 74.64±0.72 | 73.66±0.58 | 71.36±0.65 | 73.32±0.68 | 65.78±1.33 | 62.38±0.56 | 73.94±0.52 | 72.86±0.48 | 71.02±0.25 |
| | RS-GNN | 74.93±0.65 | 73.65±0.45 | 70.54±1.26 | 76.31±0.03 | 73.27±0.38 | 61.42±2.01 | 75.03±0.25 | 72.85±0.15 | 70.14±1.06 |
| | Co-teaching+ | 71.01±2.83 | 68.12±2.38 | 61.65±4.27 | 72.09±1.21 | 68.25±2.91 | 56.64±5.46 | 70.80±3.08 | 67.46±2.55 | 62.12±2.81 |
| | Me-Momentum | 75.40±0.26 | 74.41±0.56 | 70.51±0.79 | 76.93±0.47 | 74.07±1.06 | 63.96±0.97 | 75.27±0.25 | 74.24±0.45 | 71.18±0.45 |
| | MentorNet | 69.61±3.42 | 66.87±3.78 | 60.21±2.67 | 71.96±1.81 | 66.14±4.98 | 54.20±6.25 | 70.56±2.55 | 64.90±4.72 | 60.95±4.93 |
| | CLNode | 68.73±2.07 | 64.26±3.18 | 56.07±3.61 | 69.11±3.15 | 61.62±3.33 | 53.32±4.29 | 69.91±1.88 | 66.22±2.65 | 60.37±3.10 |
| | RCL | 60.90±0.12 | 54.50±2.53 | 46.58±1.44 | 65.00±0.13 | 56.68±0.27 | 51.14±1.58 | 63.70±0.53 | 54.70±1.97 | 46.62±0.59 |
| | **TCL** | **75.86±0.31** | **74.77±0.79** | **71.81±0.74** | **77.25±0.44** | **74.91±0.90** | **65.36±1.27** | **76.61±0.17** | **75.61±0.29** | **74.03±0.26** |
| PubMed | Cross-Entropy | 85.98±0.50 | 84.80±0.83 | 82.83±1.55 | 85.31±0.38 | 83.31±0.58 | 76.12±2.04 | 85.29±0.27 | 84.10±0.74 | 82.45±2.96 |
| | LPM | 85.33±0.70 | 84.33±0.79 | 82.31±0.89 | 85.90±0.57 | 84.63±0.34 | 78.94±0.79 | 85.51±0.52 | 84.90±0.53 | 83.12±1.18 |
| | CP | 86.12±0.63 | 85.01±0.65 | 82.33±1.51 | 86.13±0.36 | 84.87±0.46 | 78.81±0.77 | 85.66±0.60 | 84.92±0.99 | 81.18±1.95 |
| | NRGNN | 86.19±0.44 | 84.99±1.16 | 83.02±1.44 | 86.26±0.81 | 83.79±1.28 | 75.83±2.72 | 85.45±0.52 | 85.07±1.15 | 83.47±1.02 |
| | PI-GNN | 86.16±0.06 | 85.35±0.11 | 83.12±0.13 | 86.01±0.12 | 84.09±0.21 | 78.35±0.23 | 86.13±0.29 | 85.09±0.40 | 83.22±0.85 |
| | RT-GNN | 84.73±0.05 | 84.70±0.35 | 79.39±0.25 | 82.90±0.10 | 80.80±0.10 | 79.90±0.12 | 83.09±0.43 | 81.60±0.15 | 80.81±0.32 |
| | RS-GNN | 85.38±0.42 | 84.34±0.38 | 82.37±0.35 | 85.24±0.24 | 83.12±0.47 | 75.24±1.27 | 85.16±0.32 | 84.14±0.14 | 83.07±0.15 |
| | Co-teaching+ | 86.14±0.58 | 85.01±0.74 | 82.74±2.12 | 85.37±1.90 | 84.45±0.75 | 77.31±5.38 | 85.83±0.54 | 84.65±1.47 | 81.42±2.89 |
| | Me-Momentum | 86.05±0.18 | 85.66±0.78 | 82.42±0.41 | 85.78±0.26 | 85.43±0.35 | 80.34±0.41 | 85.87±0.27 | 84.37±0.40 | 83.53±0.14 |
| | MentorNet | 85.43±0.81 | 84.55±1.33 | 82.84±0.92 | 86.64±0.59 | 84.83±0.92 | 74.36±6.01 | 85.14±1.75 | 84.13±1.75 | 80.38±3.99 |
| | CLNode | 86.03±0.37 | 85.34±0.45 | 83.06±0.37 | 86.27±0.42 | 85.15±0.38 | 81.12±0.44 | 85.23±0.37 | 84.61±0.39 | 83.63±0.51 |
| | RCL | 82.40±0.24 | 80.30±0.15 | 76.40±0.14 | 82.70±0.23 | 82.66±0.69 | 81.30±0.20 | 82.10±0.12 | 80.30±0.12 | 74.90±0.19 |
| | **TCL** | **86.69±0.32** | **86.23±0.37** | **83.53±0.23** | **87.05±0.28** | **86.30±0.22** | **83.18±0.55** | **86.21±0.03** | **85.32±0.04** | **83.94±0.08** |

Table 2: Mean and standard deviations of classification accuracy (percentage) on large graph datasets with instance-dependent label noise. The results are the mean over five trials and the best are bolded.

| Dataset | WikiCS | | Facebook | | Physics | | DBLP | |
|---|---|---|---|---|---|---|---|---|
| Method | 30% | 50 % | 30% | 50% | 30% | 50% | 30% | 50% |
| CP | 72.27±0.40 | 54.41±1.75 | 74.86±1.19 | 62.46±3.47 | 90.64±1.38 | 81.88±0.96 | 70.02±3.06 | 55.54±5.58 |
| NRGNN | 73.09±1.63 | 56.10±2.67 | 68.00±2.34 | 58.34±3.69 | 88.96±2.23 | 82.04±1.06 | 72.48±2.61 | 65.42±9.63 |
| PI-GNN | 75.28±0.56 | 58.51±1.24 | 75.18±0.26 | 60.32±0.26 | 89.16±1.03 | 82.14±0.94 | 71.72±3.39 | 62.31±2.26 |
| Co-teaching+ | 72.64±0.81 | 54.66±2.18 | 75.19±1.53 | 60.48±3.22 | 90.08±1.71 | 78.07±4.73 | 66.32±2.12 | 51.46±4.49 |
| Me-Momentum | 75.75±0.28 | 58.40±1.95 | 62.86±1.39 | 46.13±1.67 | 82.65±0.69 | 68.22±2.47 | 59.88±0.60 | 44.54±2.34 |
| MentorNet | 72.17±0.98 | 51.80±3.30 | 73.74±2.07 | 59.04±3.38 | 88.59±2.51 | 76.31±4.50 | 63.73±4.93 | 47.85±6.47 |
| CLNode | 73.98±0.40 | 58.93±1.12 | 77.14±2.35 | 59.08±2.63 | 90.96±1.14 | 80.89±2.36 | 72.32±2.06 | 61.21±3.07 |
| RCL | 64.88±0.72 | 55.14±0.01 | 67.20±0.01 | 52.70±1.04 | 85.16±1.34 | 72.14±1.72 | 63.20±0.81 | 48.12±1.16 |
| **TCL** | **76.35±0.06** | **59.33±0.46** | **77.58±1.81** | **64.46±1.75** | **92.64±0.82** | **86.04±1.03** | **74.70±1.72** | **66.30±1.13** |

number $t$ controls the increasing speed of $\lambda$, while $\lambda_0$ controls the initial number of $\lambda$ (Wang et al., 2021b). Thus, we evaluate the sensitivity of TCL to $\lambda(t)$ and $\lambda_0$. From Fig. 5 (a), We find that the performance is relatively similar when applying different pacing functions. Additionally, the results in Fig. 5 (b) show the performance is relatively good when $\lambda_0$ is between 0.3 and 0.7.

## 3.2 ABLATION STUDY

**Performance comparison on different GNN architectures** We evaluate our proposed TCL on different GNN architectures, i.e., GCN (Zhang et al., 2019), GAT (Veličković et al., 2017), ARMA (Bianchi et al., 2021) and APPNP (Gasteiger et al., 2018). The experiments are conducted on

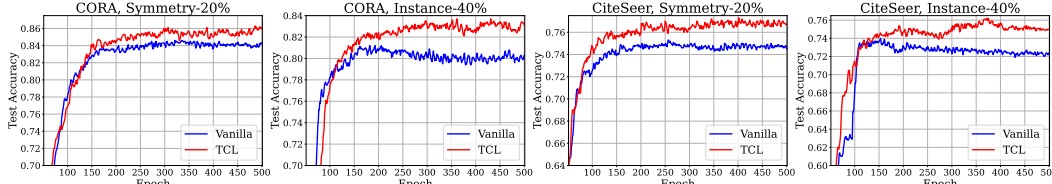

Figure 6: Illustration the effectiveness of TCL on noisy *CORA* and *CiteSeer*. "Vanilla" as a curriculum learning is based on the straightforward selection with confidence, instead of the CBC measure.

Table 3: Mean and standard deviations of classification accuracy (percentage) on different GNN architectures. The experimental results are reported over five trials. Bold numbers are superior results.

| Dataset | *CORA* | | | | *CiteSeer* | | | |
|---------|--------|--------|--------|--------|------------|--------|--------|--------|
| Backbone | *Symmetric* | | *Pairflip* | | *Symmetric* | | *Pairflip* | |
| | 20% | 40 % | 20% | 40% | 20% | 40% | 20% | 40% |
| GCN | 85.96±0.22 | 82.58±0.92 | 85.26±0.30 | 65.15±1.53 | **76.87±0.37** | 74.77±0.79 | **77.25±0.44** | **65.36±1.27** |
| GAT | 86.12±0.50 | **82.68±0.78** | 85.86±0.44 | 66.26±0.79 | 76.16±0.40 | 73.72±0.19 | 76.98±0.30 | 64.48±1.63 |
| ARMA | 85.82±0.40 | 81.32±0.80 | 84.20±0.27 | 65.48±1.11 | 75.22±0.37 | 72.80±0.60 | 75.32±0.78 | 63.86±1.17 |
| APPNP | **86.54±0.45** | 82.20±0.68 | **86.20±0.44** | **66.64±1.00** | 76.70±0.26 | **75.66±0.44** | 76.64±0.45 | 65.32±1.69 |

Table 4: Mean and standard deviations of classification accuracy (percentage) on different difficulty measurer. The experimental results are reported over five trials. Bold numbers are superior results.

| Dataset | *CiteSeer* | | | | *PubMed* | | | |
|---------|-----------|--------|--------|--------|----------|--------|--------|--------|
| Difficulty Measurer | *Symmetric* | | *Instance-dependent* | | *Symmetric* | | *Instance-dependent* | |
| | 30% | 50% | 30% | 50% | 30% | 50% | 30% | 50% |
| Feature-based | 74.35±0.86 | 68.77±0.59 | 74.50±0.16 | 70.30±0.12 | 84.11±0.76 | 81.64±0.50 | 84.10±0.04 | 81.72±0.06 |
| Neighborhood-based | 74.54±0.36 | 68.93±0.78 | 74.72±0.10 | 68.90±0.11 | 84.15±0.88 | 81.86±0.43 | 84.28±0.23 | 81.76±0.10 |
| CBC-based | **75.86±0.31** | **71.81±0.74** | **76.61±0.17** | **74.03±0.26** | **86.69±0.32** | **83.53±0.23** | **86.21±0.03** | **83.94±0.08** |

Cora and CiteSeer datasets, which are shown in Table 3. As can be seen, TCL performs similarly on different GNN architectures, showing the consistent generalization on different architectures.

**Performance comparison on different difficulty measurers**    We compare our proposed CBC measurement with other two baseline measurements: The feature-based difficulty measurer and the Neighborhood-based difficulty measurer in Table 4. The results clearly demonstrate the enhanced performance of the CBC-based difficulty measurer. Notably, the extent of accuracy improvement presents a consistent upward trend as the noise rate increases. This observation further underscores the efficacy and value of the CBC-based approach in effectively dealing with label noise.

**The underlying mechanism of TCL**    To evaluate the "easy-to-hard" mechanism of TCL, we design an *vanilla* method that extracts the confident nodes once at the beginning of training epochs and trains a GNN on the totally extracted nodes during all epochs. The initial extraction process is similar to TCL. From the comparison in the Fig. 6, we can see that the TCL gradually improves the training efficiency by introducing more confident nodes and reaches better performance than the vanilla method. This proves the necessity of introducing the "easy-to-hard" learning schedule along with CBC to alleviate the poor extraction performance from hard nodes during the cold-start stage.

## 4   CONCLUSION

To handle the challenge of extracting confident nodes on the noisily labeled graph, we propose a Class-conditional Betweenness Centrality (CBC) measure that exploits the topological information to characterize the relative difficulty of each node. With the CBC measure, we construct a novel Topological Curriculum Learning (TCL) framework that first learns high-confident nodes and then gradually introduces low-confident nodes. This "easy-to-hard" learning schedule improves training efficiency and alleviates the negative impacts of low-confident noisy nodes. The effectiveness of this framework has been proved by our theoretical analysis and extensive experiments. In the future, we will continually explore the robustness of TCL for other imperfect graph data, for example, imbalanced graph data or out-of-distribution graph data to demonstrate its effectiveness.

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

APPENDIX

## A   A FURTHER DISCUSSION ON CLASS-CONDITIONAL BETWEENNESS CENTRALITY

### A.1   BACKGROUND OF BETWEENNESS CENTRALITY

When shaping classifiers by GNNs in graph-structured data, some nodes situated near topological class boundaries are important to drive the decision boundaries of the trained classifier (Chen et al., 2021). However, GNNs find it challenging to discern class characteristics from these nodes due to their aggregation of characteristics from various classes, causing them to lack the distinctive features typical of their corresponding classes (Wei et al., 2023). Moreover, this heterogeneous aggregation makes it difficult to extract clean label nodes from those near the boundaries. Thus, we design a Class-conditional Betweenness Centrality (CBC) measure that can effectively detect those nodes.

Our Class-conditional Betweenness Centrality measure is inspired by the classical concept in graph theory – Betweenness Centrality (BC). The formal definition of the Betweenness Centrality is as follows.

**Definition A.1** (Betweenness centrality). *The betweenness centrality (BC) of the node $\mathbf{v}_i$ is defined to be the fraction of shortest paths between pairs of vertices in a graph $\mathcal{G}$ that pass through $\mathbf{v}_i$. Formally, the betweenness centrality of a node $\mathbf{v}_i$ is defined:*

$$\mathbf{b}_{\mathbf{v}_i} = \frac{1}{n(n-1)} \sum_{\mathbf{v}_u \neq \mathbf{v}_i \neq \mathbf{v}_v} \frac{\sigma_{\mathbf{v}_u,\mathbf{v}_v}(\mathbf{v}_i)}{\sigma_{\mathbf{v}_u,\mathbf{v}_v}} \tag{6}$$

*where $\sigma_{\mathbf{v}_u,\mathbf{v}_v}$ denotes the number of shortest paths from $\mathbf{v}_u$ to $\mathbf{v}_v$, and $\sigma_{\mathbf{v}_u,\mathbf{v}_v}(\mathbf{v}_i)$ denotes the number of shortest paths from $\mathbf{v}_u$ to $\mathbf{v}_v$ that pass through $\mathbf{v}_i$.*

### A.2   DIFFERENCE OF CLASS-CONDITIONAL BETWEENNESS CENTRALITY

The betweenness centrality measures the centrality of nodes in a connected graph based on the shortest paths of other pairs of nodes. It provides a quantified measure of a node's influence in controlling the flow of information among other nodes. A higher betweenness centrality signifies a node's increased significance in regulating the information flow within the network. By incorporating the class-conditional constraint into Eq. (6), we can effectively identify nodes that play a crucial role in controlling the flow of information between different classes and are typically located near topological class boundaries. This is exemplified by the boundary-near nodes $v_5$ and $v_9$ in Fig. 7, where the shortest paths for nodes in class 1 and class 2 must pass through these nodes, underlining their pivotal role in managing information flow between the two classes.

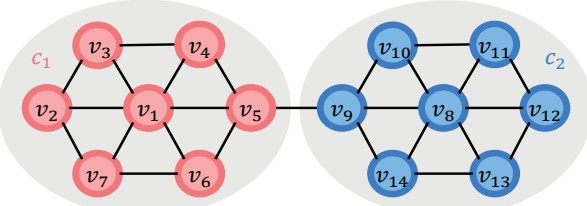

Figure 7: The illustration of boundary-near nodes

Thus, after adding the class-conditional constrain into the Eq.(6), we define the CBC of a node $\mathbf{v}_i$ as the fraction of shortest paths between pairs of nodes that belong to different classes in a graph $\mathcal{G}$ that pass through $\mathbf{v}_i$:

$$\mathbf{Cb}_i = \frac{1}{n(n-1)} \sum_{\substack{\mathbf{v}_u \neq \mathbf{v}_i \neq \mathbf{v}_v \\ y_u \neq y_v}} \frac{\sigma_{\mathbf{v}_u,\mathbf{v}_v}(\mathbf{v}_i)}{\sigma_{\mathbf{v}_u,\mathbf{v}_v}} \tag{7}$$

where $\sigma_{\mathbf{v}_u,\mathbf{v}_v}$ denotes the number of shortest paths from $\mathbf{v}_u$ to $\mathbf{v}_v$, and $\sigma_{\mathbf{v}_u,\mathbf{v}_v}(\mathbf{v}_i)$ denotes the number of shortest paths from $\mathbf{v}_u$ to $\mathbf{v}_v$ that pass through $\mathbf{v}_i$.

Notably, the CBC measure builds upon the BC measure and outperforms it in detecting boundary-near nodes. This improvement is attributed to the class-conditional constraint, which alleviates the impact

of information flow among nodes belonging to the same class. Specifically, information flow among nodes of the same class is more likely to occur through nodes positioned near the centre of the class rather than at the boundary. For instance, in Fig. 7, the shortest path from node $v_3$ to $v_6$ or from $v_7$ to $v_4$ traverses the centre-near node $v_1$ rather than the boundary-near node $v_5$.

### A.3 OPTIMIZATION FORM OF CLASS-CONDITIONAL BETWEENNESS CENTRALITY

However, it is usually practically limited to directly employ Eq. (7), since in most networks, the information does not flow only along the shortest paths (Stephenson and Zelen, 1989; Freeman et al., 1991; Newman, 2005), and it is very time-consuming to find the shortest paths in a large graph (Liu and Lü, 2010; Zhao et al., 2022). Thus, we relax Eq. (7) with the *random walk*, which simultaneously allows the multiple paths to contribute to CBC and avoids the expensive search cost of the shortest paths (Noh and Rieger, 2004; Liu and Lü, 2010; Zhao et al., 2022). Concretely, we employ the Personalized PageRank (PPR) method (Bahmani et al., 2010; Haveliwala et al., 2003) to implement random walk and then arrive at the final form of our CBC in the following definition.

**Definition A.2** (Class-conditional Betweenness Centrality). *Given the Personalized PageRank matrix $\boldsymbol{\pi} = \alpha(\mathbf{I} - (1-\alpha)\hat{\mathbf{A}})^{-1}$ ($\boldsymbol{\pi} \in \mathbb{R}^{n \times n}$), the Class-conditional Betweenness Centrality of the node $\mathbf{v}_i$ is defined by counting how often the node $\mathbf{v}_i$ is traversed by a random walk between pairs of other vertices that belong to different classes in a graph $\mathcal{G}$:*

$$\mathbf{Cb}_i := \frac{1}{n(n-1)} \sum_{\substack{\mathbf{v}_u \neq \mathbf{v}_i \neq \mathbf{v}_v \\ \tilde{y}_u \neq \tilde{y}_v}} \frac{\boldsymbol{\pi}_{u,i}\boldsymbol{\pi}_{i,v}}{\boldsymbol{\pi}_{u,v}}, \quad (8)$$

*where $\boldsymbol{\pi}_{u,i}$ with the target node $\mathbf{v}_u$ and the source node $\mathbf{v}_i$ denotes the probability that an $\alpha$-discounted random walk from node $\mathbf{v}_u$ terminates at $\mathbf{v}_i$. Here an $\alpha$-discounted random walk represents a random traversal that, at each step, either terminates at the current node with probability $\alpha$, or moves to a random out-neighbour with probability $1 - \alpha$.*

In the above definition, the CBC is based on the random walks that count how often a node is traversed by a random walk between pairs of other nodes that belong to different classes. Our proposed CBC successfully detects the boundary-near nodes by evaluating the flow of messages passing between different classes. The nodes that possess high CBC are closer to the topological class boundaries. Consequently, our CBC measure is adept at identifying the topological structure of nodes, and its exploration of topological information renders it robust against noisy labeled data. Additionally, the CBC measure can be seamlessly integrated into other related domains. For instance, it can be employed to identify the structure of nodes in out-of-distribution (OOD) detection tasks, as discussed in Wu et al. (2022), and to enhance OOD generalization, as demonstrated in studies by Yang et al. (2022) and Wu et al. (2021).

### A.4 IMPORTANCE OF CLASS-CONDITIONAL BETWEENNESS CENTRALITY

In Fig. 8, we present a visual representation highlighting the clear positive correlation between test accuracy and the aggregate Class-conditional Betweenness Centrality (CBC) of the training set. Additionally, we carefully structure the training sequence for each node in every training set, prioritizing nodes based on their CBC scores. This underlines the pivotal role of CBC in shaping the performance of models. The empirical findings strongly affirm the significance of extracting insights from informative nodes, a factor that markedly enhances the performace of GNNs throughout the training process.

### A.5 DISTRIBUTION OF CLASS-CONDITIONAL BETWEENNESS CENTRALITY

In our comprehensive empirical analysis, we thoroughly investigate the distributions of Class-conditional Betweenness Centrality for nodes in WikiCS, considering diverse levels of noise as presented in Fig. 9. To pre-categorize nodes based on their proximity to topological class boundaries, we employ the following criteria: Nodes are classified as "far from topological class boundaries" (far from boundaries) if their two-hop neighbors belong to the same class. Conversely, nodes are labeled as "near topological class boundaries" (near boundaries) if this condition does not apply. It's important to note that the "WikiCS" dataset, chosen for this analysis, is substantial and comprises sparsely

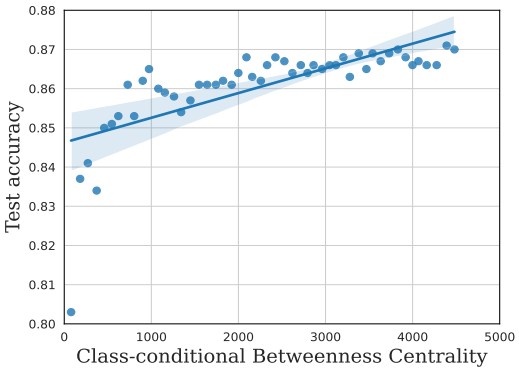

Figure 8: There is a significant positive correlation between the test accuracy and the overall CBC of the clean labeled training set (the Pearson correlation coefficient is 0.6999 over 50 randomly selected class-balanced training sets with the $p$ value smaller than 0.0001).

labeled nodes. As observed in Fig. 9, the node dataset exhibits two distinct clusters. Even in the presence of considerable label noise, nodes far away from topological class boundaries consistently demonstrate lower CBC scores across all cases.

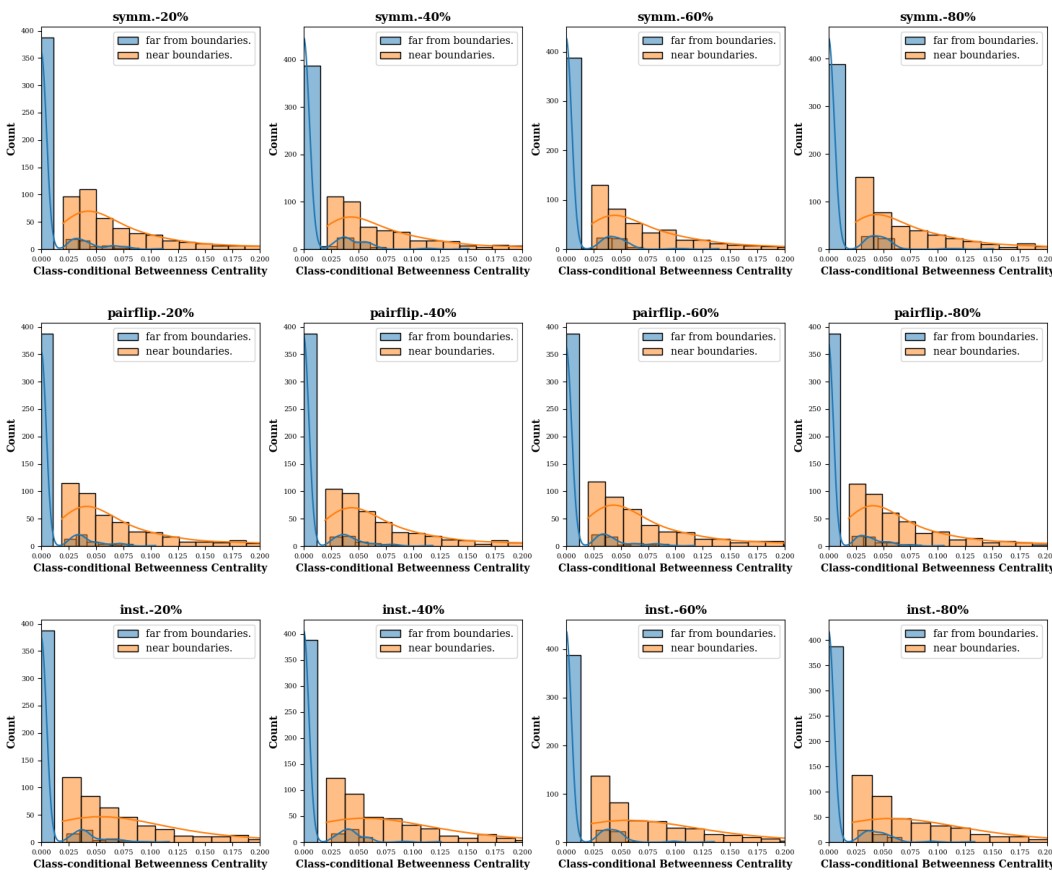

Figure 9: Class-conditional Betweenness Centrality distributions of nodes in WikiCS, with varying levels of symmetric noise (symm.), pairflip noise (pairflip.), and instance-based noise (inst.).

# B   RELATED WORK

## B.1   CURRICULUM LEARNING WITH LABEL NOISE

We have diligently incorporated curriculum-based approaches into our literature review that align with our research theme. One widely adopted criterion involves selecting samples with small losses and treating them as clean data. Several curriculum learning methods utilize this criterion (Jiang et al., 2014), and in each step, select samples with small losses. For instance, in MentorNet (Jiang et al., 2018), an additional pre-trained network is employed to select clean instances using loss values to guide model training. The underlying concept of MentorNet resembles the self-training approach (Kumar et al., 2010), inheriting the drawback of accumulated error due to sample-selection bias.

To address this issue, Co-teaching (Han et al., 2018) and Co-teaching+ (Yu et al., 2019) mitigate the problem by training two DNNs and using the loss computed on one model to guide the other. CurriculumNet (Guo et al., 2018)presents a curriculum learning approach based on unsupervised estimation of data complexity through its distribution in a feature space. It benefits from training with both clean and noisy samples and weights each sample's loss in training based on the gradient directions compared to those on validation (*i.e*, , a clean set). Notably, CurriculumNet relies on a clean validation set.

It's worth emphasizing that the discussed curriculum learning methods primarily focus on mitigating label noise issues within i.i.d. datasets and depend on the prediction of pre-trained neural networks. However, those methods cannot be employed on graph data due to the "over-smoothing" issue when training Graph Neural Networks (GNNs). Note that, in GNNs, "over-smoothing" refers to the phenomenon where, as the network depth increases, node features become increasingly similar. This similarity poses a challenge when employing curriculum learning with label noise, making it difficult to distinguish between "easy" and "hard" nodes due to the homogenization of features caused by over-smoothing. Additionally, even in shallow GNN architectures, over-smoothing can lead to under-confident predictions, complicating the task of establishing an 'easy-to-hard' training curriculum (Wang et al., 2021a; Hsu et al., 2022). Addressing this challenge, our work introduces a novel method, which proposes a robust CBC measure. This measure effectively distinguishes between 'easy' and 'hard' nodes, taking into account the graph structure rather than the prediction of GNNs, thereby mitigating the over-smoothing problem. Our work stands as a pioneer in the development of a curriculum learning approach explicitly designed for graph data afflicted by label noise. This distinction underscores a significant contribution of our research, emphasizing the necessity for specialized strategies to effectively handle noise within graph-structured data.

## B.2   GRAPH NEURAL NETWOEKS

Predicting node labels involves formulating a parameterized hypothesis using the function $f_{\mathcal{G}}(\mathbf{A}, \mathcal{X}) = \hat{y}_{\mathbf{A}}$, incorporating a Graph Neural Network (GNN) architecture (Kipf and Welling, 2016) and a message propagation framework (Gilmer et al., 2017). The GNN architecture can take on various forms such as GCN (Kipf and Welling, 2016), GAT (Veličković et al., 2017), or GraphSAGE (Hamilton et al., 2017).

In practical terms, the forward inference of an $L$-layer GNN involves generating node representations $\boldsymbol{H}_{\mathbf{A}} \in \mathbb{R}^{N \times D}$ through $L$-layer message propagation. Specifically, with $\ell = 1 \dots L$ denoting the layer index, $h_i^{\ell}$ is the representation of the node $i$, MESS$(\cdot)$ being a learnable mapping function to transform the input feature, AGGREGATE$(\cdot)$ capturing 1-hop information from the neighborhood $\mathcal{N}(v)$ in the graph, and COMBINE$(\cdot)$ signifying the final combination of neighbor features and the node itself, the $L$-layer operation of GNNs can be formulated as $\boldsymbol{m}_v^{\ell} = \text{AGGREGATE}^{\ell}(\{\text{MESS}(\boldsymbol{h}_u^{\ell-1}, \boldsymbol{h}_v^{\ell-1}, e_{uv}) : u \in \mathcal{N}(v)\})$, where $\boldsymbol{h}_v^{\ell} = \text{COMBINE}^{\ell}(\boldsymbol{h}_v^{\ell-1}, \boldsymbol{m}_v^{\ell})$. After L-layer propagation, the final node representations $\boldsymbol{h}_e^{L}$ for each $e \in V$ are derived. Furthermore, a detailed summary of different GNN architectures is presented in Table 5.

Subsequently, a subsequent linear layer transforms $\boldsymbol{H}_{\mathbf{A}}$ into classification probabilities $\hat{y}_{\mathbf{A}} \in \mathbb{R}^{N \times C}$, where $C$ represents the total categories. The primary training objective is to minimize the classification loss, typically measured by cross-entropy between the predicted $\hat{y}_{\mathbf{A}}$ and the ground truth $Y$.

Table 5: Detailed architectures of different GNNs.

| GNN | MESS($\cdot$) & AGGREGATE($\cdot$) | COMBINE($\cdot$) |
|---|---|---|
| GCN | $\boldsymbol{m}_i^l = \boldsymbol{W}^l \sum_{j \in \mathcal{N}(i)} \frac{1}{\sqrt{\hat{d}_i \hat{d}_j}} \boldsymbol{h}_j^{l-1}$ | $\boldsymbol{h}_i^l = \sigma(\boldsymbol{m}_i^l + \boldsymbol{W}^l \frac{1}{\hat{d}_i} \boldsymbol{h}_i^{l-1})$ |
| GAT | $\boldsymbol{m}_i^l = \sum_{j \in \mathcal{N}(i)} \alpha_{ij} \boldsymbol{W}^l \boldsymbol{h}_j^{l-1}$ | $\boldsymbol{h}_i^l = \sigma(\boldsymbol{m}_i^l + \boldsymbol{W}^l \alpha_{ii} \boldsymbol{h}_i^{l-1})$ |
| GraphSAGE | $\boldsymbol{m}_i^l = \boldsymbol{W}^l \frac{1}{|\mathcal{N}(i)|} \sum_{j \in \mathcal{N}(i)} \boldsymbol{h}_j^{l-1}$ | $\boldsymbol{h}_i^l = \sigma(\boldsymbol{m}_i^l + \boldsymbol{W}^l \boldsymbol{h}_i^{l-1})$ |

### B.3 DENOISING METHODS ON GRAPH DATA

Prior research has explored diverse strategies to address the challenge of label noise in graph data. NRGNN (Dai et al., 2021) combats label noise by linking unlabeled nodes with noisily labeled nodes that share high feature similarity, thus incorporating more reliable label information. Conversely, PI-GNN (Du et al., 2021) mitigates noise impact by introducing Pairwise Intersection (PI) labels based on feature similarity among nodes.

In a different approach, the LPM method (Xia et al., 2020a) and GNN-Cleaner (Xia et al., 2023) address noisy labels by involving a small set of clean nodes for assistance. Additionally, CP (Zhang et al., 2020) operates with class labels derived from clustering node embeddings, encouraging the classifier to capture class-cluster information and avoid overfitting to noisy labels.

Furthermore, RS-GNN (Dai et al., 2022) focuses on enhancing GNNs' robustness to noisy edges. It achieves this by training a link predictor on noisy graphs, aiming to enable effective learning from graphs that contain inaccuracies in edge connections.

Lastly, RT-GNN (Qian et al., 2023) leverages the memorization effect of neural networks to select clean labeled nodes, generating pseudo-labels from these selected nodes to mitigate the influence of noisy nodes on the training process.

In addition, the efficacy of graph contrastive learning has been harnessed to effectively reduce label noise during node classification tasks on graph-based data. Based on the homophily assumption, ALEX (Yuan et al., 2023a) learns robust node representations utilizing graph contrastive learning to mitigate the overfitting of noisy nodes and CGNN (Yuan et al., 2023b) integrates graph contrastive learning as a regularization term, thereby bolstering the robustness of trained models against label noise. Each of these approaches offers unique insights into effectively handling label noise in graph data.

In this context, our proposed Topological Curriculum Learning (TCL) represents a distinctive perspective on employing curriculum learning methods specifically tailored for noisily labeled graphs. By introducing TCL, we contribute a novel and effective strategy to tackle label noise in the complex domain of graph-structured data.

### B.4 GRAPH CURRICULUM LEARNING

Graph Curriculum Learning (GCL) stands at the intersection of graph machine learning and curriculum learning, gaining increasing prominence due to its potential. At its core, GCL revolves around customizing a difficulty measure to compute a difficulty score for each data sample, crucial in defining an effective learning curriculum for the model. The design of this difficulty measure can follow predefined or automatic approaches.

Predefined approaches often employ heuristic metrics to measure node difficulty based on specific characteristics even before the training commences. For example, CLNode (Wei et al., 2023) gauges node difficulty by considering label diversity among a node's neighbors. Conversely, SMMCL (Gong et al., 2019) assumes varying difficulty levels among different samples for propagation, advocating an easy-to-hard sequence in the curriculum for label propagation.

On the other hand, automatic approaches determine difficulty during training using a supervised learning paradigm rather than predefined heuristic-based metrics. For example, RCL (Zhang et al., 2023) gradually incorporates the relation between nodes into training based on the relation's difficulty, measured using a supervised learning approach. Another instance, MentorGNN (Zhou et al., 2022),

tailors complex graph signals by deriving a curriculum for pre-training GNNs to learn informative node representations and enhance generalization performances.

However, a notable gap exists in existing GCL methods concerning their robustness to label noise, especially in effectively handling graphs with noisy labels. Our proposed Topological Curriculum Learning (TCL) addresses this limitation by being the pioneer in curriculum learning explicitly designed for graphs affected by label noise. This underscores the novelty and significance of TCL within the domain of GCL.

## C  PROOF TO THEORETICAL GUARANTEE OF TCL

### C.1  PROOF FOR THE WEIGHTED EXPRESSION

We first formulate $\mathbb{P}_{\mathcal{D}}(\mathbf{A}, \mathbf{x})$ as the weighted expression of $\mathbb{P}_{\hat{\mathcal{D}}}(\mathbf{A}, \mathbf{x})$:

$$\mathbb{P}_{\mathcal{D}}(\mathbf{A}, \mathbf{x}) = \frac{1}{\alpha^*} W_{\lambda^*}(\mathbf{A}, \mathbf{x}) \mathbb{P}_{\hat{\mathcal{D}}}(\mathbf{A}, \mathbf{x}), \tag{9}$$

where $0 \leq W_{\lambda^*}(\mathbf{A}, x) \leq 1$ and $\alpha^* = \int_{\mathbf{A}, \mathcal{X}} W_{\lambda^*}(\mathbf{A}, \mathbf{x}) \mathbb{P}_{\hat{\mathcal{D}}}(\mathbf{A}, \mathbf{x}) dx$ denote the normalization factor. Based on Eq.(9), $\mathbb{P}_{\mathcal{D}}(\mathbf{A}, \mathbf{x})$ actually corresponding to a curriculum as definied in Eq.(2) under the weight function $W_{\lambda^*}(\mathbf{A}, \mathbf{x})$.

Eq.(9) can be equivalently reformulated as

$$\mathbb{P}_{\hat{\mathcal{D}}}(\mathbf{A}, \mathbf{x}) = \alpha^* \mathbb{P}_{\mathcal{D}}(\mathbf{A}, \mathbf{x}) + (1 - \alpha^*) E(\mathbf{A}, \mathbf{x}),$$

where

$$E(\mathbf{A}, \mathbf{x}) = \frac{1}{1 - \alpha^*} (1 - W_{\lambda^*}(\mathbf{A}, \mathbf{x})) \mathbb{P}_{\hat{\mathcal{D}}}(\mathbf{A}, \mathbf{x}).$$

Here, the term $E(\mathbf{A}, \mathbf{x})$ measures the deviation from $\mathbb{P}_{\hat{\mathcal{D}}}(\mathbf{A}, \mathbf{x})$ to $\mathbb{P}_{\mathcal{D}}(\mathbf{A}, \mathbf{x})$. Recalling the previous empirical analysis of Fig. 4, extracting confident nodes from the early $\tilde{Q}_\lambda$ that emphasises the easy nodes works well. We define this period (corresponding to relatively small $\lambda$) as the high-confidence regions. In these high-confidence areas, $\mathbb{P}_{\mathcal{D}}(\mathbf{A}, \mathbf{x})$ is accordant to the $\mathbb{P}_{\hat{\mathcal{D}}}(\mathbf{A}, \mathbf{x})$ and thus $E(\mathbf{A}, \mathbf{x})$ corresponds to the nearly zero-weighted $\mathbb{P}_{\hat{\mathcal{D}}}(\mathbf{A}, \mathbf{x})$ tending to be small. On the contrary, in later training criteria, the poor performance of extracting confident nodes causes that the $\mathbb{P}_{\hat{\mathcal{D}}}(\mathbf{A}, \mathbf{x})$ cannot approximate the $\mathbb{P}_{\hat{\mathcal{D}}}(\mathbf{A}, \mathbf{x})$ well in those low-confident regions. $E(\mathbf{A}, \mathbf{x})$ then imposes large weights on $\mathbb{P}_{\hat{\mathcal{D}}}(\mathbf{A}, \mathbf{x})$, yielding the large deviation values. Combining with Definition 2.2, we construct the below curriculum sequence for theoretical evaluation:

$$\hat{Q}_\lambda(\mathbf{A}, \mathbf{x}) \propto W_\lambda(\mathbf{A}, \mathbf{x}) \mathbb{P}_{\hat{\mathcal{D}}}(\mathbf{A}, \mathbf{x}), \tag{10}$$

where

$$W_\lambda(\mathbf{A}, \mathbf{x}) \propto \frac{\alpha_\lambda \mathbb{P}_{\mathcal{D}}(\mathbf{A}, \mathbf{x}) + (1 - \alpha_\lambda) E(\mathbf{A}, \mathbf{x})}{\alpha^* \mathbb{P}_{\mathcal{D}}(\mathbf{A}, \mathbf{x}) + (1 - \alpha)^* E(\mathbf{A}, \mathbf{x})}$$

with $0 \leq W_\lambda(\mathbf{A}, \mathbf{x}) \leq 1$ through normalizing its maximal value as 1 and $\alpha_\lambda$ varies from 1 to $\alpha^*$ with increasing pace parameter $\lambda$.

### C.2  PROOF OF THEOREM 1

Now, we estimate the expected risk by the following surrogate (Donini et al., 2018):

$$\mathcal{R}_{\mathrm{emp}}(f_{\mathcal{G}}) := \frac{1}{n} \sum_{i=1}^{n_{\mathrm{cf}}} \mathcal{L}_{f_{\mathcal{G}}}(z_i). \tag{11}$$

Let $\mathcal{F}$ be a function family mapping from $Z_{\mathbf{x}_i}$ to $[a, b]$, $\mathbb{P}(Z_{\mathbf{x}_i})$ a distribution on $Z_{\mathbf{x}_i}$ and $S = (Z_{\mathbf{x}_1}, \dots, Z_{\mathbf{x}_m})$ a set of i.i.d. samples drawn from $\mathbb{P}$. The empirical Rademacher complexity of $\mathcal{F}$ with respect to $S$ is defined by

$$\hat{\mathfrak{R}}_m(\mathcal{F}) = \mathbb{E}_\sigma [\sup_{g \in \mathcal{F}} \frac{1}{m} \sum_{i=1}^m \sigma_i g(Z_{\mathbf{x}_i})], \tag{12}$$

where $\sigma_i$ are i.i.d. samples drawn from the uniform distribution in $\{-1, 1\}$. The Rademacher complexity of $\mathcal{F}$ is defined by the expectation of $\hat{\mathfrak{R}}_m$ over all samples $S$:

$$\mathfrak{R}_m(\mathcal{F}) = \mathbb{E}_{S \sim \mathbb{P}^m} |\hat{\mathfrak{R}}_S(\mathcal{F})|. \tag{13}$$

**Definition C.1.** *The Kullback-Leibler divergence $D_{KL}(p||q)$ between two densities $p(\Omega)$ and $q(\Omega)$ is defined by*

$$D_{KL}(p||q) = \int_\Omega p(\mathbf{x}) \log \frac{p(\mathbf{x})}{q(\mathbf{x})} d\mathbf{x}. \tag{14}$$

Based on the above definitions, we can estimate the generalization error bound for curriculum learning under the curriculum $\hat{Q}_\lambda$. Based on the Bretagnolle-Huber inequality (Schlüter et al., 2013), we have

$$\int |p(\mathbf{x}) - q(\mathbf{x})| d\mathbf{x} \leq 2\sqrt{1 - \exp\{-D_{KL}(p||q)\}} \tag{15}$$

Let $\mathcal{H}$ be a family of functions taking value in $\{-1, 1\}$, for any $\delta > 0$ with confidence at least $1 - \delta$ over a sample set $S$, the following holds for any $f_{\mathcal{G}} \in \mathcal{H}$ (Gong et al., 2016):

$$\mathcal{R}(f_{\mathcal{G}}) \leq \mathcal{R}_{emp}(f_{\mathcal{G}}) + \mathfrak{R}_m(\mathcal{H}) + \sqrt{\frac{\ln(\frac{1}{\delta})}{2m}}. \tag{16}$$

In addition, we have

$$\mathcal{R}(f_{\mathcal{G}}) \leq \mathcal{R}_{emp}(f_{\mathcal{G}}) + \mathfrak{R}_m(\mathcal{H}) + 3\sqrt{\frac{\ln(\frac{1}{\delta})}{2m}}. \tag{17}$$

Suppose $S \subseteq \{\mathbf{x} : \|\mathbf{x}\| \leq R\}$ be a sample set of size $m$, and $\mathcal{H} = \{x \to sgn(\mathbf{w}^T \mathbf{x}) : min_s |\mathbf{w}^T \mathbf{x}| = 1 \cap \|\mathbf{w}\| \leq B\}$ be hypothesis class, where $\mathbf{w} \in \mathbb{R}^n, \mathbf{x} \in \mathbb{R}^n$, and then we have

$$\hat{\mathfrak{R}}_m(\mathcal{H}) \leq \frac{BR}{\sqrt{m}} \tag{18}$$

*Proof.*

$$\begin{aligned}
\hat{\mathfrak{R}}_m(\mathcal{H}) &= \frac{1}{m}\mathbb{E}_\sigma\left[\sup_{\|\mathbf{w}\| \leq B} \sum_{i=1}^m \sigma_i sgn(\mathbf{w}\mathbf{x}_i)\right] \\
&\leq \frac{1}{m}\mathbb{E}_\sigma\left[\sup_{\|\mathbf{w}\| \leq B} \sum_{i=1}^m \sigma_i |sgn(\mathbf{w}\mathbf{x}_i)|\right] \leq \frac{1}{m}\mathbb{E}_\sigma\left[\sup_{\|\mathbf{w}\| \leq B} \sum_{i=1}^m \sigma_i |\mathbf{w}\mathbf{x}_i|\right] \\
&\leq \frac{B}{m}\mathbb{E}_\sigma\left[\sum_{i=1}^m \sigma_i \|\mathbf{x}_i\|\right] \leq \frac{B}{m}\mathbb{E}_\sigma\left[|\sum_{i=1}^m \sigma_i \|\mathbf{x}_i\||\right] \\
&= \frac{B}{m}\mathbb{E}_\sigma\left[\sqrt{(\sum_{i=1}^m \sigma_i \|\mathbf{x}_i\|)^2}\right] \\
&= \frac{B}{m}\mathbb{E}_\sigma\left[\sqrt{\sum_{i,j=1}^m \sigma_i \sigma_j \|\mathbf{x}_i\| \|\mathbf{x}_j\|}\right] \\
&\leq \frac{B}{m}\sqrt{\mathbb{E}_\sigma\left[\sum_{i,j=1}^m \sigma_i \sigma_j \|\mathbf{x}_i\| \|\mathbf{x}_j\|\right]} \\
&= \frac{B}{m}\sqrt{\sum_{i=1}^m \|\mathbf{x}_i\|^2} \\
&\leq \frac{BR}{\sqrt{m}}.
\end{aligned} \tag{19}$$

$\square$

Then, suppose $\{(Z_{\mathbf{x}_i}, y_i)\}_{i=1}^m$ are i.i.d. samples drawn from the confident pace distribution $\hat{Q}_\lambda$. Denote $m_+/m_-$ be the number of positive/negative samples and $m^* = \min\{m_-, m_+\}$. $\mathcal{H}$ is the function family projecting to $\{-1, 1\}$. Then for any $\delta > 0$ and $f \in \mathcal{H}$, with confidence at least $1 - 2\delta$ we have:

$$
\begin{aligned}
\mathcal{R}(f_\mathcal{G}) \leq\ & \frac{1}{2}\mathcal{R}_{emp}^+(f_\mathcal{G}) + \frac{1}{2}\mathcal{R}_{emp}^-(f_\mathcal{G}) \\
& + \frac{1}{2}\hat{\mathfrak{R}}_{m_+}(\mathcal{H}) + \frac{1}{2}\hat{\mathfrak{R}}_{m_-}(\mathcal{H}) + \sqrt{\frac{\ln(\frac{2}{\delta})}{m^*}} \\
& + (1 - \alpha_\lambda)\sqrt{1 - \exp\{-D_{KL}(\mathbb{P}_\mathcal{D}^+\|E^+)\}} \\
& + (1 - \alpha_\lambda)\sqrt{1 - \exp\{-D_{KL}(\mathbb{P}_\mathcal{D}^-\|E^-)\}},
\end{aligned}
\tag{20}
$$

and

$$
\begin{aligned}
\mathcal{R}(f_\mathcal{G}) \leq\ & \frac{1}{2}\mathcal{R}_{emp}^+(f_\mathcal{G}) + \frac{1}{2}\mathcal{R}_{emp}^-(f_\mathcal{G}) \\
& + \frac{1}{2}\hat{\mathfrak{R}}_{m_+}(\mathcal{H}) + \frac{1}{2}\hat{\mathfrak{R}}_{m_-}(\mathcal{H}) + 3\sqrt{\frac{\ln(\frac{2}{\delta})}{m^*}} \\
& + (1 - \alpha_\lambda)\sqrt{1 - \exp\{-D_{KL}(\mathbb{P}_\mathcal{D}^+\|E^+)\}} \\
& + (1 - \alpha_\lambda)\sqrt{1 - \exp\{-D_{KL}(\mathbb{P}_\mathcal{D}^-\|E^-)\}},
\end{aligned}
\tag{21}
$$

where $E^+, E^-$ denotes the error distribution corresponding to $\mathbb{P}_\mathcal{D}(\mathbf{A}, x|y = 1), \mathbb{P}_\mathcal{D}(\mathbf{A}, x|y = -1)$, and $\mathcal{R}_{emp}^+(f_\mathcal{G}), \mathcal{R}_{emp}^-(f_\mathcal{G})$ denote the empirical risk on positive nodes and negative nodes, respectively.

*Proof.* We first rewrite the expected risk as:

$$
\begin{aligned}
\mathcal{R}(f_\mathcal{G}) &= \int_Z \mathcal{L}_{f_\mathcal{G}}(z)\mathbb{P}_\mathcal{D}(\mathbf{A}, \mathbf{x}|y)\mathbb{P}_\mathcal{D}(y)dz, \\
&= \frac{1}{2}\int_{\mathcal{X}^+} \mathcal{L}_{f_\mathcal{G}}(\mathbf{x}, y)\mathbb{P}_\mathcal{D}(\mathbf{A}, \mathbf{x}|y = 1)dx + \frac{1}{2}\int_{\mathcal{X}^-} \mathcal{L}_{f_\mathcal{G}}(x, y)\mathbb{P}_\mathcal{D}(\mathbf{A}, \mathbf{x}|y = -1)dx \\
&:= \frac{1}{2}(\mathcal{R}^+(f_\mathcal{G}) + \mathcal{R}^-(f_\mathcal{G})).
\end{aligned}
\tag{22}
$$

The empirical risk tends not to approximate the expected risk due to the inconsistency of $\mathbb{P}_{\hat{\mathcal{D}}}(\mathbf{A}, \mathbf{x}|y)$ and $\mathbb{P}_\mathcal{D}(\mathbf{A}, \mathbf{x}|y)$. However, by introducing the error distribution with the confident pace distribution and denoting by $\mathbb{E}_{\hat{Q}_\lambda}(f_\mathcal{G})$ in the error analysis, we can the following error decomposition:

$$
\begin{aligned}
& \frac{1}{2}(\mathcal{R}^+(f_\mathcal{G}) + \mathcal{R}^-(f_\mathcal{G})) - \frac{1}{2}(\mathcal{R}_{emp}^+(f_\mathcal{G}) + \mathcal{R}_{emp}^-(f_\mathcal{G})) \\
&= \frac{1}{2}[\mathcal{R}^+(f_\mathcal{G}) - \mathbb{E}_{\hat{Q}_\lambda^+}(f_\mathcal{G}) + \mathbb{E}_{\hat{Q}_\lambda^+}(f_\mathcal{G}) - \mathcal{R}_{emp}^+(f_\mathcal{G})] \\
&+ \frac{1}{2}[\mathcal{R}^-(f_\mathcal{G}) - \mathbb{E}_{\hat{Q}_\lambda^-}(f_\mathcal{G}) + \mathbb{E}_{\hat{Q}_\lambda^-}(f_\mathcal{G}) - \mathcal{R}_{emp}^-(f_\mathcal{G})] \\
&:= S_1 + S_2.
\end{aligned}
\tag{23}
$$

Let $S_1 = A_1 + A_2$ and $S_2 = B_1 + B_2$, where $A_1 = \frac{1}{2}(\mathcal{R}^+(f_\mathcal{G})) - \mathbb{E}_{\hat{Q}_\lambda^+}(f_\mathcal{G})$, $A_2 = \frac{1}{2}(\mathbb{E}_{\hat{Q}_\lambda^+}(f_\mathcal{G}) - \mathcal{R}_{emp}^+(f_\mathcal{G}))$, $B_1 = \frac{1}{2}(\mathcal{R}^-(f_\mathcal{G})) - \mathbb{E}_{\hat{Q}_\lambda^-}(f_\mathcal{G})$, $B_2 = \frac{1}{2}(\mathbb{E}_{\hat{Q}_\lambda^-}(f_\mathcal{G}) - \mathcal{R}_{emp}^-(f_\mathcal{G}))$. Here, $\mathbb{E}_{\hat{Q}_\lambda^+}(f_\mathcal{G})$ and $\mathbb{E}_{\hat{Q}_\lambda^-}(f_\mathcal{G})$ denote the pace risk with respect to positive nodes and negative nodes, respectively.

By the fact, the 0-1 loss is bounded by 1, we have:

$$
\begin{aligned}
A_1 + A_2 =& \frac{1}{2}[\mathcal{R}^+(f_\mathcal{G}) - \mathbb{E}_{\hat{Q}_\lambda^+}(f_\mathcal{G}) + \mathbb{E}_{\hat{Q}_\lambda^+}(f_\mathcal{G}) - \mathcal{R}_{emp}^+(f_\mathcal{G})] \\
\leq& \frac{1}{2}\int_{\mathcal{X}_+} (\mathbb{P}_\mathcal{D}(\mathbf{A}, x|y) - \hat{Q}_\lambda^+(x))dx + \frac{1}{2}\mathfrak{R}_{m_+}(\mathcal{H}) + \frac{1}{2}\sqrt{\frac{\ln(\frac{1}{\delta})}{2m_+}} \\
\leq& (1-\alpha_\lambda)\sqrt{1 - \exp\left\{-D_{KL}(\mathbb{P}_\mathcal{D}^+||E^+)\right\}} + \frac{1}{2}\mathfrak{R}_{m_+}(\mathcal{H}) + \frac{1}{2}\sqrt{\frac{\ln(\frac{1}{\delta})}{2m_+}}.
\end{aligned}
\tag{24}
$$

In a similar way, we can bound:

$$
\begin{aligned}
B_1 + B_2 =& \frac{1}{2}[\mathcal{R}^-(f_\mathcal{G}) - \mathbb{E}_{\hat{Q}_\lambda^-}(f_\mathcal{G}) + \mathbb{E}_{\hat{Q}_\lambda^-}(f_\mathcal{G}) - \mathcal{R}_{emp}^-(f_\mathcal{G})] \\
\leq& (1-\alpha_\lambda)\sqrt{1 - \exp\left\{-D_{KL}(\mathbb{P}_\mathcal{D}^-||E^-)\right\}} + \frac{1}{2}\mathfrak{R}_{m_-}(\mathcal{H}) + \frac{1}{2}\sqrt{\frac{\ln(\frac{1}{\delta})}{2m_-}}.
\end{aligned}
\tag{25}
$$

By taking $m^* = \min\{m_-, m_+\}$ and combine Eq. (24) and Eq. (25), we can get:

$$
\begin{aligned}
\mathcal{R}(f_\mathcal{G}) \leq& \frac{1}{2}\mathcal{R}_{emp}^+(f_\mathcal{G}) + \frac{1}{2}\mathcal{R}_{emp}^-(f_\mathcal{G}) \\
&+ \frac{1}{2}\hat{\mathfrak{R}}_{m_+}(\mathcal{H}) + \frac{1}{2}\hat{\mathfrak{R}}_{m_-}(\mathcal{H}) + \sqrt{\frac{\ln(\frac{2}{\delta})}{m^*}} \\
&+ (1-\alpha_\lambda)\sqrt{1 - \exp\left\{-D_{KL}(\mathbb{P}_\mathcal{D}^+||E^+)\right\}} \\
&+ (1-\alpha_\lambda)\sqrt{1 - \exp\left\{-D_{KL}(\mathbb{P}_\mathcal{D}^-||E^-)\right\}}.
\end{aligned}
\tag{26}
$$

In addition, we further get:

$$
\mathfrak{R}_m(\mathcal{H}) \leq \hat{\mathfrak{R}}_m(\mathcal{H}) + \sqrt{\frac{\ln(\frac{2}{\delta})}{2m}}.
\tag{27}
$$

By replacing $\mathfrak{R}_m$, we complete the proof. $\qquad\square$

The above established error bounds upon 0-1 loss are hard to optimize. We change the bound of Eq.(21) under the commonly utilized hinge loss $\phi(t) = (1-t)_+$ for $t \in \mathbb{R}$ and finally obtain our Theorem 1. The above proof is according to (Gong et al., 2016).

## D    DETAILS OF EMPIRICAL STUDY

### D.1    DATASETS

In our experiments, we employ seven common datasets gathered from diverse domains. The datasets are as follows: (1) Cora, CiteSeer, and Pubmed (Yang et al., 2016), which are citation networks where nodes represent documents and edges signify citations among them; (2) WikiCS (Mernyei and Cangea, 2020), comprising nodes corresponding to Computer Science articles. Edges are based on hyperlinks, and the ten classes represent different branches of the field in the Wikipedia website; (3) Facebook (Rozemberczki et al., 2021), with nodes representing verified pages on Facebook and edges indicating mutual likes; (4) Physics (Shchur et al., 2018), a co-authorship graph based on the Microsoft Academic Graph. In this dataset, nodes represent authors connected by an edge if they co-authored a paper. Node features represent paper keywords for each author's papers, and class labels indicate the most active fields of study for each author; (5) DBLP (Pan et al., 2016), also a citation network, where each paper may cite or be cited by other papers. The statistical information for the utilized datasets is presented in Table 6.

Table 6: Important statistical information of used datasets.

| Dataset | Edges | Classes | Features | Nodes/Labeled Nodes | Labeled Ratio |
|---------|-------|---------|----------|---------------------|---------------|
| Cora | $5,429$ | 7 | $1,433$ | $2,708/1208$ | 44.61% |
| CiteSeer | $4,732$ | 6 | $3,703$ | $3,327/1827$ | 54.91% |
| PubMed | $44,338$ | 3 | 500 | $19,717/18217$ | 92.39% |
| WikiCS | $215,603$ | 10 | 300 | $11,701/580$ | 4.96% |
| Facebook | $342,004$ | 4 | 128 | $22,470/400$ | 1.78% |
| Physics | $495,924$ | 5 | 8415 | $34,493/500$ | 1.45% |
| DBLP | $105,734$ | 4 | 1639 | $17,716/800$ | 4.51% |

## D.2 LABEL NOISE GENERATION SETTING

Following previous works (Dai et al., 2021; Du et al., 2021; Xia et al., 2020b), we consider three settings of simulated noisy labels:

(1) *Symmetric* noise: this kind of label noise is generated by flipping labels in each class uniformly to incorrect labels of other classes.

(2) *Pairflip* noise: the noise flips each class to its adjacent class. More explanation about this noise setting can be found in (Yu et al., 2019; Zheng et al., 2020; Lyu and Tsang, 2019).

(3) *Instance-dependent* noise: the noise is quite realistic, where the probability that an instance is mislabeled depends on its features. We follow (Xia et al., 2020b) to generate this type of label noise to validate the effectiveness of the proposed method.

## D.3 BASELINE DETAILS

In more detail, we employ baselines:

- *Curriculum learning with label noise on i.i.d. data:*
    (1) Co-teaching+ (Yu et al., 2019): This approach employs a dual-network mechanism to reciprocally extract confident samples. Specifically, instances with minimal loss and discordant predictions are identified as reliable, clean samples for subsequent training.
    (2) Me-Momentum (Bai and Liu, 2021): The objective of this method is to identify challenging clean examples from noisy training data. This process involves iteratively updating the extracted examples while refining the classifier.
    (3) MentorNet (Jiang et al., 2018): This approach involves pre-training an additional network, which is then used to select clean instances and guide the training of the main network. In cases where clean validation data is unavailable, the self-paced variant of MentorNet resorts to a predefined curriculum, such as focusing on instances with small losses.

- *Graph Curriculum learning:*
    (1) CLNode (Wei et al., 2023): CLNode is a curriculum learning framework aimed at enhancing the performance of backbone GNNs by gradually introducing more challenging nodes during the training process. The proposed difficulty measure is based on label information.
    (2) RCL (Zhang et al., 2023): RCL utilizes diverse underlying data dependencies to train improved Graph Neural Networks (GNNs), resulting in enhanced quality of learned node representations. It gauges the inter-node relationships as a measure of difficulty for each node.

- *Denoising methods on graph data:*
    (1) LPM (Xia et al., 2020a): The method is specifically tailored to address noisy labels in node classification, employing a small set of clean nodes for guidance.
    (2) CP (Zhang et al., 2020): The method operates on class labels derived from clustering node embeddings. It encourages the classifier to comprehend class-cluster information, effectively mitigating overfitting to noisy labels. Prior to clustering, node embeddings are acquired using the Node2Vec model (Grover and Leskovec, 2016).

(3) NRGNN (Dai et al., 2021): In this approach, a label noise-resistant GNN establishes connections between unlabeled nodes and noisily labeled nodes with high feature similarity. This connection strategy effectively incorporates additional clean label information into the model.

(4) PI-GNN (Du et al., 2023): This method introduces Pairwise Intersection (PI) labels, generated based on feature similarity among nodes. These PI labels are then employed to alleviate the adverse impact of label noise, thereby enhancing the model's robustness.

(6) RS-GNN (Dai et al., 2022) This method primarily aims to improve the robustness of Graph Neural Networks (GNNs) in the presence of noisy edges. It achieves this by training a link predictor on graphs with inaccuracies in edge connections, ultimately enabling GNNs to effectively learn from such imperfect graph structures.

(5) RT-GNN (Qian et al., 2023): This approach identifies clean labeled nodes by leveraging the memorization effect of neural networks. Subsequently, it generates pseudo-labels based on these selected clean nodes to mitigate the impact of noisy nodes during the training process.

## D.4 ALGORITHM FRAMEWORK OF TCL

---

**Algorithm 1** Algorithm flow of TCL.

---

**Input:** A pretrained classifier $f_{\mathcal{G}}^p$, the noisy training set $\tilde{\mathcal{D}}_{\mathrm{tr}} = \{(\mathbf{A}, x_i, \tilde{y}_i)\}_{i=1}^{n_{\mathrm{tr}}}$, the identity matrix $\mathbf{I}$, the normalized adjacency matrix $\hat{\mathbf{A}}$, the hyperparameters $\alpha, \lambda_0, T$.

**Output:** The trained GNN classifier $f_{\mathcal{G}}$.

1 Obtain $\boldsymbol{\pi} \leftarrow \alpha(\mathbf{I} - (1-\alpha)\hat{\mathbf{A}})^{-1}$ ;

2 Initialize parameters of a GNN classifier $f_{\mathcal{G}}$;

3 **for** $\mathbf{v}_i \in \tilde{\mathcal{D}}_{\mathrm{tr}}$ **do**

4 $\quad$ Calculate $\mathbf{Cb}_i \leftarrow \frac{1}{n_{\mathrm{tr}}(n_{\mathrm{tr}}-1)} \sum_{\substack{\mathbf{v}_u \neq \mathbf{v}_i \neq \mathbf{v}_v \\ \tilde{y}_u \neq \tilde{y}_v}} \frac{\boldsymbol{\pi}_{u,i}\boldsymbol{\pi}_{i,v}}{\boldsymbol{\pi}_{u,v}},$

5 **end**

6 Sort $\tilde{\mathcal{D}}_{\mathrm{tr}}$ according to $\mathbf{Cb}_i$ in ascending order;

7 Let $t = 1$;

8 **while** $t < T$ *or not converge* **do**

9 $\quad \lambda_t \leftarrow \min(1, \lambda_{t-1} + (1-\lambda_{t-1}) * \frac{t}{T})$;

10 $\quad$ Generate noisy training subset $\tilde{\mathcal{D}}_{\mathrm{tr}}^t \leftarrow \tilde{\mathcal{D}}_{\mathrm{tr}}[1, \ldots, \lfloor \lambda_t * n_{\mathrm{tr}} \rfloor]$;

11 $\quad$ Extract confident training subset $\hat{\mathcal{D}}_{\mathrm{tr}}^t$ from $\tilde{\mathcal{D}}_{\mathrm{tr}}^t$;

$\quad$ // i.e., the training nodes whose noisy labels are identical to the ones predicted by $f_{\mathcal{G}}^p$

12 $\quad$ Calculate loss $\mathcal{L}$ on $\hat{\mathcal{D}}_{\mathrm{tr}}^t$;

13 $\quad$ Back-propagation on $f_{\mathcal{G}}$ for minimizing $\mathcal{L}$;

14 $\quad t \leftarrow t + 1$;

15 **end**

---

## D.5 PACING FUNCTION OF TCL

After measuring node difficulty using the CBC measure, we employ the TCL framework to enhance the training of our GNN model. We incorporate a pacing function $\lambda(t)$ to govern the proportion $\lambda$ of training nodes available at the $t$-th epoch. In TCL, we utilize three distinct pacing functions: linear, root, and geometric.

- linear:

$$\lambda_t = \min(1, \lambda_{t-1} + (1-\lambda_{t-1}) * \frac{t}{T}) \tag{28}$$

- root:

$$\lambda_t = \min(1, \sqrt{\lambda_{t-1}^2 + (1-\lambda_{t-1}^2) * \frac{t}{T}}) \tag{29}$$

- geometric:

$$\lambda_t = \min(1, 2^{\log_2 \lambda_t - \log_2 \lambda_t * \frac{t}{T}}) \tag{30}$$

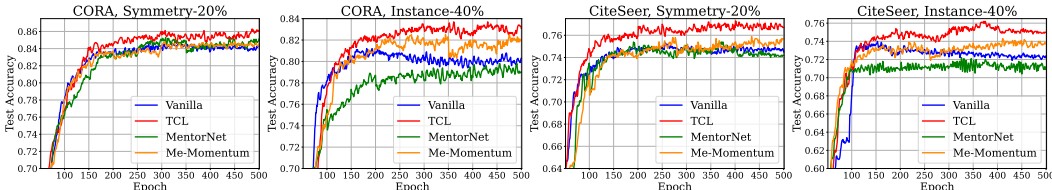

Figure 10: Illustration the effectiveness of TCL on noisy *CORA* and *CiteSeer*.

The linear function escalates the training node difficulty uniformly over epochs. On the other hand, the root function introduces a higher proportion of difficult nodes in a smaller number of epochs. Meanwhile, the geometric function extends the training duration on a subset of easy nodes by conducting multiple epochs.

### D.6 REPRODUCTION ENVIRONMENT

We run all the experiments on a Linux server, some important information is listed:

- CPU: AMD EPYC 7302 16-Core Processor $\times$ 64
- GPU: NVIDIA GeForce RTX4090
- RAM: 188GB
- cuda: 11.6

**Python Package**  We implement all deep learning methods based on Python 3.8. The experiment code is included in the supplementary materials. The versions of some important packages are listed:

- torch: 1.11.0
- torch-geometric: 2.0.4
- numpy:1.21.4
- scipy:1.8.1

### D.7 IMPLEMENTATION DETAILS

A two-layer graph convolutional network whose hidden dimension is 16 is deployed as the backbone for all methods. We apply an Adam optimizer (Kingma and Ba, 2014) with a learning rate of $0.01$. The weight decay is set to $5 \times 10^{-4}$. The number of pre-training epochs is set to 400. While the number of retraining epochs is set to 500 for Cora, CiteSeer, and 1000 for Pubmed, WikiCS, Facebook, Physics and DBLP. All hyper-parameters are tuned based on a noisy validation set built by leaving 10% noisy training data.

## E MORE EXPERIMENT

### E.1 COMPLETED EXPERIMENT OF THE UNDERLYING MECHANISM OF TCL

To evaluate the "easy-to-hard" mechanism of TCL, we design an *vanilla* method that extracts the confident nodes once at the beginning of training epochs and trains a GNN on the totally extracted nodes during all epochs. The initial extraction process is similar to TCL. From the comparison in the Fig. 10, we can see that the TCL gradually improves the training efficiency by introducing more confident nodes and reaches better performance than the vanilla method. Additionally, the utilization of two baseline curriculum learning methods further demonstrates the effectiveness of our approach. This proves the necessity of introducing the "easy-to-hard" learning schedule along with CBC to alleviate the poor extraction performance from hard nodes during the cold-start stage.

Table 7: Mean and standard deviations of classification accuracy (percentage) on heterphily graph datasets with 30% instance-dependent label noise. The results are the mean over five trials and the best are bolded.

| Method | *Chameleon* | *Squirrel* | *DBLP* |
|---|---|---|---|
| CP | 55.08±2.18 | 43.42±2.46 | 70.02±3.06 |
| NRGNN | 49.02±2.35 | 41.35±1.98 | 72.48±2.61 |
| PI-GNN | 52.85±2.16 | 43.31±2.97 | 71.72±3.39 |
| Co-teaching+ | 53.07±1.98 | 39.48±2.54 | 66.32±2.12 |
| Me-Momentum | 55.01±1.69 | 44.38±1.78 | 59.88±0.60 |
| MentorNet | 53.73±3.75 | 39.63±3.43 | 63.73±4.93 |
| CLNode | 52.85±2.91 | 35.92±1.84 | 72.32±2.06 |
| RCL | 52.96±0.96 | 40.59±1.23 | 63.20±0.81 |
| TCL | **56.17±0.28** | **48.03±1.03** | **74.70±1.72** |

## E.2 CBC DISTRIBUTIONS OF NODES WITH VARYING HOMOPHILY RATIO

In this section, we assess the effectiveness of our CBC measure in relation to varying homophily ratios within the noisy labeled graph. We modify the graph structure by introducing synthetic, cross-label (heterophilous) edges that connect nodes with differing labels. The methodology for adding these heterophilous edges, as well as the calculation for the homophily ratio, are both referred to (Ma et al., 2021). As illustrated in Fig. 11, a decrease in the homophily ratio results in an increased number of nodes near class boundaries, which consequently exhibit higher CBC scores. Notably, our CBC measure effectively reflects the topology of nodes even as the complexity of the graph increases.

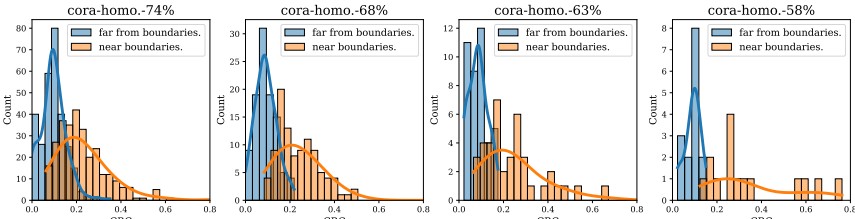

Figure 11: The distributions of the CBC score *w.r.t.* nodes on CORA with different homophily ratios in the presence of 30% instance-dependent label noise. The nodes are considered "far from topological class boundaries" (far from boundaries.) when their two-hop neighbours belong to the same class; conversely, nodes are categorized as "near topological class boundaries" (near boundaries.) when this condition does not hold.

## E.3 PERFORMANCE COMPARISON ON HETERPHILY DATASETS

We evaluate the effectiveness of our method on three commonly used heterogeneous datasets, i.e., DBLP (Fu et al., 2020), Chameleon (Rozemberczki et al., 2021), Squirrel (Rozemberczki et al., 2021) under 30% instance-dependent label noise. The summary of experimental results is in the Table 7. As can be seen, our method still shows superior performance over a range of baselines.

## F LIMITATIONS

Indeed, our TCL method has demonstrated effectiveness across various scenarios. However, it's important to acknowledge certain inherent limitations due to the intricacies of dealing with noisily labeled graphs.

Firstly, the TCL method is specifically tailored for homogeneously-connected graphs, where linked nodes are anticipated to share similarities. This is evident in the diverse datasets utilized in our experiments. Adapting TCL to heterogeneously connected graphs, such as protein networks, requires a nuanced refinement of our approach to suit the distinct network characteristics.

Secondly, a notable challenge for TCL arises when the labeling ratio is exceptionally low. In such instances, the extraction of clean nodes might inadvertently overlook crucial features of mislabeled nodes. This oversight could potentially impact the learning process of models. Addressing this limitation mandates thoughtful adjustments in our approach, aiming to accommodate scenarios with scantily labeled data better.

## G  BROADER IMPACTS

Noisy labels have become prevalent in the era of big data, posing significant reliability challenges for traditional supervised learning algorithms. The impact of label noise is even more pronounced in graph data, where the noise can propagate through topological edges. Effectively addressing noisy labels on graph data is a critical issue that significantly impacts the practical applications of graph data, garnering increasing attention from both the research and industry communities.

In this study, we introduce a Topological Curriculum Learning (TCL) framework to mitigate the adverse effects of label noise by selectively extracting nodes with clean labels. The effectiveness of TCL is supported by substantial evidence detailed in the paper. The outcomes of this research will advance our understanding of handling label noise in graph data and substantially enhance the robustness of graph models, making strides toward more reliable and accurate graph-based learning.

