# OpenReview forum: "Mitigating Label Noise on Graphs via Topological Curriculum Learning"
_ICLR.cc/2024/Conference — Submitted to ICLR 2024_

### Official Review · Reviewer_gbE5 · 2023-10-15

**Soundness:** 2 fair
**Presentation:** 2 fair
**Contribution:** 2 fair
**Rating:** 3
**Confidence:** 3

**Summary:**

This paper studies the problem of robust learning on graphs and proposes a new selection criteria named CBC, which are incorprated into a curriculum learning framework to select confident nodes. Authors experimentally show the superiority of our method compared with state-of-the-art baselines.

**Strengths:**

1. The motivation of the work is sound.

2. The paper is well written.

3. The paper is well organized.

**Weaknesses:**

1. The incorporation of curriculum learning to graph robust learning has been studied in [1,2,3]. Although there are some differences, [1] can smoothly adapted into the studied problem.

2. The performance compared with the best baseline is minor in a lot of datasets, e.g., PubMed of less than 1%, which is even bigger than the variance with different GNN architectures in Table 3.

3. The performance with naive curriculum learning (selection with confidence) should be included in Figure 6.

4. The introduction of CBC to topological curriculum learning is not clear. I expect to see the selection criteria more clear. However, authors show the Eqn. 2 with abstract W_\lamdba. It should be a very simple selection criteria as from Algorithm in Appendix.

[1] OMG: Towards Effective Graph Classification Against Label Noise, TKDE 2023.

[2] Curriculum Graph Machine Learning: A Survey, IJCAI 2023.

[3] CuCo: Graph Representation with Curriculum Contrastive Learning., IJCAI 2021.

**Questions:**

See weakness.

---

> ### Author Response · Authors · 2023-11-19
> **The response to Reviewer  gbE5 (Part 1)**
>
> We would like to first thank Reviewer gbE5 for carefully reading our submission and we carefully considered your comments please find the answers to your questions below.
>
> > w1:
>
> We have thoroughly reviewed listed papers [1, 2, 3], finding that papers [1] and [3] focus on graph classification tasks, which **significantly differ** from our study’s focus on node classification. In graph classification, each graph is treated as an independent entity with a single label, akin to **i.i.d datasets**. Conversely, node classification deals with interdependent labeled nodes within a single interconnected graph, is **not a i.i.d dataset structure**. Therefore, methods suited for graph classification **cannot smoothly transfer** to node classification issues, due to their fundamental distinction within the graph machine learning community. The paper [2] provides a thorough survey of curriculum graph machine learning, but it **does not include** methods for implementing curriculum learning to tackle label noise in node classification task. We have already discussed this point on our **related work (Appendix B.4)**.
>
> We appreciate the reviewer's recommendation, and for more discussion about the distinction from these works [1, 2, 3], please refer to the **revised Related Work (Appendix B)** section in the updated submission.
>
> [1] OMG: Towards Effective Graph Classification Against Label Noise, TKDE 2023.
>
> [2] Curriculum Graph Machine Learning: A Survey, IJCAI 2023.
>
> [3] CuCo: Graph Representation with Curriculum Contrastive Learning., IJCAI 2021.
>
> > w2:
>
> The term "minor" indeed encompasses a broad assertion. First, it's crucial to highlight that in every cases, our approach **consistently outperforms** existing curriculum learning (CL) methods in terms of average accuracy. Second, **as the second-best baselines change in case-by-case noise levels, it is not reasonable to roughly consider the comparison between RCL with the case-by-case second-best baseline**. It is more fair to compare the **average improvement** between TCL and other baselines across all cases within a single dataset. These calculations, based on the data from our Tables 1 and 2, are **summarized in the table below**. The data presented in the table below illustrates the superiority of our method over the second-best approaches, with an significant improvement margin in all datasets. Notably, **while the second-best methods are different across various datasets, our method consistently emerges the top performance.** This evidence demonstrates that our approach substantially outperforms other baseline methods and is effective on noisy labeled graph.
>
>
> **The average improvement of the following table is calulated by averaging the accuracy improvement between TCL and other methods across all cases.** For example, as we can see, although Me-Momentum is competitive, TCL achieves 6.52\%, 9.91\%, and 18.29\% improvement on average on Facebook, Physics and DBLP datasets.
>
> ||  CORA   | CiteSeer  | Pubmed | WiKiCS | Facebook | Physics | DBLP |
> |----|  ----  | ----  | ----  | ----  | ----  | ----  |----  |
> |CP| 4.34  | 5.68 |1.49|4.51|**2.36**|**3.08**|7.72|
> |NRGNN| 3.11  |  **1.28**| 1.60|3.25|7.85|3.84|**1.55**|
> |PI-GNN| 2.40 |4.14|1.21|1.05|3.27|3.69|3.48|
> |Co-teaching+|4.18 |7.56|1.73|4.19|3.18|5.27|11.61|
> |Me-Momentum |**1.34** | 1.31|1.15|**0.97**|6.52|9.91|18.29|
> |MentorNet |5.11 | 8.98 |2.24|5.85|4.63|6.89|14.71|
> |CLNode  |6.22|10.73|**1.09** |1.38|2.91|3.42|3.74|
> |RCL  |15.83 |18.49|5.04|7.83|11.07|10.69|14.84|
>
> In Tables 1 and 2, where we compare accuracy, we ensure consistency in the backbone model of our work with other baselines to maintain a fair comparison. Hence, the variance across different GNN architectures cannot be considered as a factor to influence the accruacy comparsion between our method and baseline methods.

---

> ### Author Response · Authors · 2023-11-19
> **The response to Reviewer gbE5 (Part 2)**
>
> > w3:
>
> Actually, **the "Vanilla" method in Figure 6 is the naive curriculum learning** that selects nodes according to the confidence during training. We have highlighted this point in the revised caption of Figure 6.
>
> Additionlly, to further address the concren, we have include two additional curriculum leanring method, e.g. MentorNet and Me-momentum, which use the curriculum learning to address label noise but do not employ the CBC measure. These two methods could be considered as other curriculum learning method without the aid of CBC measure. The updated experiment are now included into **the Figure 10 in Appendix E.1**. The results further demonstrate the effectiveness of our proposed CBC measure.
>
> > w4:
>
> In the curriculum learning literatrue [1, 2], the key is a measure that can estimate the learning difficulty of each instance. Following the similar spirit, our CBC is worked as a difficulty measurer in the topological curriculum learning. Specifically, the $W_\lambda$ in Eqn.2 is a weight function for each confident nodes, which is defined by our CBC measurement. This formalization has been widely used in the related curriculum learning literature [1, 2]. With the help of the CBC measure, we can design an robust "easy-to-hard" learning curriculum, akin to the approach outlined in **lines 3-6 of Algorithm 1 in Appendix D.3**.
>
> To address the reviewer's concern, we have **enriched the discussion in the corresponding parts to improve the clarity of CBC for topological curriculum learning** in the revised submission. Thank you very much for the suggestion.
>
> [1] Bengio, Y., Louradour, J., Collobert, R., & Weston, J. (2009, June). Curriculum learning. In Proceedings of the 26th annual international conference on machine learning (pp. 41-48).
>
> [2] Wang, X., Chen, Y., & Zhu, W. (2021). A survey on curriculum learning. IEEE Transactions on Pattern Analysis and Machine Intelligence, 44(9), 4555-4576.

---

> ### Author Response · Authors · 2023-11-21
>
> Dear reviewers gbE5,
>
> We have diligently addressed most of your concerns. If you still have concerns about our paper, please join the rolling discussion, where we await your valuable insights and comments.
>
> Thanks.

---

> ### Author Response · Authors · 2023-11-22
> **Looking forward to your reply**
>
> Dear Reviewer gbE5:
>
> Our sincere thanks go out to you for reviewing this paper! We do our best to address the issues raised by relevant work, experimental results and more empirical evidence. The discussion deadline is approaching now. Is there anything unclear in these explanations and descriptions?
>
> It would be greatly appreciated if your concerns had been addressed. However, if you require any further clarification, we can provide it before the deadline for discussion.
>
> Thanks!
>
> Authors

---

> > ### Comment · Reviewer_gbE5 · 2023-11-23
> >
> > Thanks for your comment. I checked the general response. R2 comment "However, it does not theoretically or, more shallowly, logically demonstrate or explain the connection between the proposed CBC criterion and noise labels." while in general response it says "supported by theoretical results (R1,R2,R3) ".

---

> > > ### Author Response · Authors · 2023-11-23
> > > **Response to Reviewer gbE5**
> > >
> > > Dear Reviewer gbE5,
> > >
> > > In the part of "summary" in the R2 comments, it is said that "This paper introduces Class-conditional Betweenness Centrality (CBC) as a robust measure to address graph label noise and develop a Topological Curriculum Learning (TCL) framework guided by CBC, which **enhances** model learning and outperforms existing methods both **theoretically** and experimentally." Thus, we could summary this point in our general response and says "supported by theoretical results".
> > >
> > > Furthermore, we are keen to address any additional concerns you might have regarding our rebuttal content or any other aspect of our work.
> > >
> > > Best regards,
> > >
> > > Authors

---

> ### Comment · Reviewer_gbE5 · 2023-11-23
>
> Thanks for your comment. However, it seems what R2's really rate about "theoretical results" is the other one and I suggested to revise the general response. Moreover, I checked the other's comment with soundness and performance, and decided to keep the rating.

---

### Official Review · Reviewer_GKC8 · 2023-10-28

**Soundness:** 3 good
**Presentation:** 3 good
**Contribution:** 3 good
**Rating:** 6
**Confidence:** 3

**Summary:**

This paper study the problem of node classification when the graph data is noisily labeled. The authors introduce the class-conditional betweenness centrality as the measure of the confidence of a node having noise label, and then apply the curriculum learning framework to guide model learning. They further show that this framework minimizes an upper bound of the expected risk under target clean distribution. Experiments on benchmark datasets demonstrate the effective of their proposed method.

**Strengths:**

- The proposed method is novel and reasonable, which incorporates the graph structural information to judge the confidence of nodes. The theoretical results also provide a suitable explanation for the effectiveness of TCL.
- The experimental results are substantial and credible, which is sufficient to support the effectiveness of TCL.

**Weaknesses:**

- It seems that there exists some error in the proof of Theorem 1 (See Questions for more detail).
- The proposed framework relies on the homogeneity of graph data, which could limit its applications in real-world scenarios, where the connected nodes may belong to different classes.

**Questions:**

1. In Eq. (18), the authors use the following inequality:
$$
\frac{1}{m} \mathbb{E}_{\sigma} \left[ \mathop{\rm sup}\_{\Vert \mathbf{w} \Vert \leq B } \sum\_{i=1}^m \sigma_i \vert \mathbf{w} \mathbf{x}_i \vert \right] \leq \frac{B}{m} \mathbb{E}\_{\sigma} \left[ \mathop{\rm sup}\_{\Vert \mathbf{w} \Vert \leq B} \Vert \sum\_{i=1}^m \sigma_i \mathbf{x}_i \Vert \vert \right].
$$

I don't think this inequality holds true. To remedy this, I recommend the following proof:
$$
\begin{aligned}
& \frac{1}{m} \mathbb{E}_{\sigma} \left[ \mathop{\rm sup}\_{\Vert \mathbf{w} \Vert \leq B } \sum\_{i=1}^m \sigma_i \vert \mathbf{w} \mathbf{x}_i \vert \right] \\\\
\leq & \frac{B}{m} \mathbb{E}\_{\sigma} \left[  \sum\_{i=1}^m \sigma_i \Vert \mathbf{x}_i \Vert \right] \leq \frac{B}{m} \mathbb{E}\_{\sigma} \left[ \left\vert  \sum\_{i=1}^m \sigma_i \Vert \mathbf{x}_i \Vert \right\vert \right] \\\\
= &  \frac{B}{m} \mathbb{E}\_{\sigma} \left[ \sqrt{\left( \sum\_{i=1}^m \sigma_i \Vert \mathbf{x}_i \Vert \right)^2} \right] =  \frac{B}{m} \mathbb{E}\_{\sigma} \left[ \sqrt{ \sum\_{i,j=1}^m \sigma_i \sigma_j \Vert \mathbf{x}_i \Vert \Vert \mathbf{x}_j \Vert} \right] \\\\
\leq & \frac{B}{m} \sqrt{ \mathbb{E}\_{\sigma}  \left[ \sum\_{i,j=1}^m \sigma_i \sigma_j \Vert \mathbf{x}_i \Vert \Vert \mathbf{x}_j \Vert\right] } = \frac{B}{m} \sqrt{ \sum\_{i=1}^m \Vert \mathbf{x}_i \Vert^2} \leq \frac{BR}{\sqrt{m}}.
\end{aligned}
$$
Also, in the first equation, the term $sgn(\mathbf{w}_i \mathbf{x}_i)$ should be corrected as $sgn(\mathbf{w} \mathbf{x}_i)$.

2. The noise label situation has a close relation with the out-of-distribution (OOD) situation. Could the proposed framework be applied to OOD generalization [1] or detection [2] tasks?

3. In Algorithm 1, why the authors use a pretrained classifier $f^p_{\mathcal{G}}$? What pretrain method was used to obtain $f^p_{\mathcal{G}}$? Do different pretrain methods affect the final performance of the model?

[1] Wu et al, Geometric knowledge distillation for topological shifts. NeurIPS 2022.

[2] Wu et al, Energy-based detection model for OOD nodes on graphs. ICLR 2023.

---

> ### Author Response · Authors · 2023-11-19
> **The response to Reviewer GKC8**
>
> We sincerely thank you for taking the time to review our manuscript and providing us with your insightful feedback. We carefully considered your comments and please find the answers to your questions below.
>
> > w1 & Q1:
>
> Thank you very much for your pretty detailed advice and we already updated our proof in **Appendix C.2** following your advice.
>
> > w2:
>
> We would like to kindly argue that our proposed framework **still remains effective on heterogeneous graphs**, as even for heterogeneous graphs, the proposed CBC measure can still efficiently distinguish the easy nodes and hard nodes. To address the reviewer's concern, we have included additional toy experiments in **Fig. 11 (Appendix E.2)**. These experiments empirically demonstrate the **effectiveness of our CBC measure in graphs with different homophily ratios**, reinforcing its utility in heterogeneous graph. Besides, we have extended our experimentation to include three commonly used heterogeneous datasets (the DBLP dataset has been included in our experiment) under 30% instance-dependendent label noise. The summary of experimental results in the below table. As can be seen, our method still show the superior performance over a range of baselines. Additionally, it's important to note that the recent research has already demonstrated the capability of GNN to handle heterogeneous graphs [1,2].
>
> |dataset | DBLP   | Chameleon | Squirrel |
> |----|  ----  | ----  | ----  |
> |CP|70.02 | 55.08| 43.42|
> |NRGNN|72.48 |49.02 | 41.35|
> |PI-GNN|71.72 |52.85 |43.31 |
> |Co-teaching+| 66.32| 53.07 | 39.48 |
> |Me-Momentum |59.88  |55.01 | 44.38 |
> |MentorNet | 63.73 |53.73 |39.63 |
> |CLNode  | 72.32|52.85 |35.92 |
> |RCL  | 63.20|52.96 | 40.59 |
> |**TCL**| **74.70**|  **56.17** | **48.03** |
>
> [1] Ma, Y., Liu, X., Shah, N., & Tang, J. Is Homophily a Necessity for Graph Neural Networks?. ICLR 2021.
>
> [2] Luan, S., Hua, C., Xu, M., Lu, Q., Zhu, J., Chang, X. W., ... & Precup, D. (2023). When do graph neural networks help with node classification: Investigating the homophily principle on node distinguishability. NIPS 2023
>
> > Q2:
>
> We appreciate about the reviewer's question. We guess that the proposed CBC measure may also benefit to the OOD scenarions, as the noise can be intrinsically considered as the OOD data. We **have included extra discussion about the potential connection with the OOD generalization [1] or detection [2], which we place in Appendix A.3** of the revised submission.
>
> [1] Wu et al, Geometric knowledge distillation for topological shifts. NeurIPS 2022.
>
> [2] Wu et al, Energy-based detection model for OOD nodes on graphs. ICLR 2023.
>
> > Q3:
>
> The pretrained classifier is used to extract the confident nodes from the noisy labeled graph and we obtain a pertrained GNN classifier based on the memorization effect of neural networks. This is because the GNN classifier pre-trained at early epochs would fit the clean data well but not the incorrectly labeled data. Thus, this pre-trained model serves as an efficient tool for identifying confident nodes within the graph. This part has been detailed illustrated in the part of **section 2.3**.
>
> Different pretraining methods do not significantly influence the performance of our model. This is primarily because the essence of our TCL framework lies in the implementation of the "easy-to-hard" curriculum learning mechanism (the Fig.6). This principle has been empirically substantiated in our extensive ablation study. Consequently, the role of pretraining methods is confined to rigorously extracting confident nodes, thereby ensuring that they do not substantially affect the final outcomes of our TCL framework.

---

> ### Author Response · Authors · 2023-11-21
>
> Dear reviewers GKC8,
>
> We have diligently addressed most of your concerns. If you still have concerns about our paper, please join the rolling discussion, where we await your valuable insights and comments.
>
> Thanks.

---

> ### Author Response · Authors · 2023-11-22
> **Looking forward to your reply**
>
> Dear Reviewer GKC8:
>
> Our sincere thanks go out to you for reviewing this paper! We do our best to address the issues raised by relevant work, experimental results and more empirical evidence. The discussion deadline is approaching now. Is there anything unclear in these explanations and descriptions?
>
> It would be greatly appreciated if your concerns had been addressed. However, if you require any further clarification, we can provide it before the deadline for discussion.
>
> Thanks!
>
> Authors

---

> ### Comment · Reviewer_GKC8 · 2023-11-22
> **Response to the Authors**
>
> Thanks the authors for their responses. My concerns are well addressed. Currently, I would like to keep my initial score.

---

### Official Review · Reviewer_cJzR · 2023-10-28

**Soundness:** 2 fair
**Presentation:** 3 good
**Contribution:** 2 fair
**Rating:** 5
**Confidence:** 4

**Summary:**

This paper introduces Class-conditional Betweenness Centrality (CBC) as a robust measure to address graph label noise and develop a Topological Curriculum Learning (TCL) framework guided by CBC, which enhances model learning and outperforms existing methods both theoretically and experimentally.

**Strengths:**

The authors propose a novel metric for assessing the cleanliness of labels, and the paper is clear, understandable, with extensive experiments

**Weaknesses:**

However, I have some concerns about certain aspects of the paper:
- This paper seems to address the problem of noisy labels on graphs with the concept of curriculum learning and Pagerank. However, it does not theoretically or, more shallowly, logically demonstrate or explain the connection between the proposed CBC criterion and noise labels. Alternatively, it does not clearly state the rationality of the CBC criterion to eliminate the negative effect of noisy labels. It is insufficient to show the correlation between CBC and class boundary nodes under the setting of noisy labels. Moreover, empirical results on a single dataset are not enough to support some conclusive statements in the paper.
- Theorem 1 draws the conclusion from the i.i.d data but not the graph-structured data directly. It is better to show a more explicit upper bound from the graph itself.
- Please explain the meaning of positive/negative samples and error distribution in Theorem 1. In addition, the upper bound relates to the clean distribution, but not the noisy distribution. Is it independent of the noise factor? If so, how can Theorem 1 be used to illustrate the connection between the proposed method and noisy labels?
- Some of the latest papers on label noise in graphs haven't been adequately mentioned and compared, which weakens the persuasiveness of the paper: (1) ALEX: Towards Effective Graph Transfer Learning with Noisy Labels. MM'23 (2) Learning on Graphs under Label Noise. ICASSP'23 (3) GNN Cleaner- Label Cleaner for Graph Structured Data. TKDE'23

**Questions:**

See Weaknesses.

---

> ### Author Response · Authors · 2023-11-19
> **The response to Reviewer cJzR (Part 1)**
>
> We sincerely thank you for taking the time to review our manuscript and providing us with your insightful feedback. We carefully considered your comments and please find the answers to your questions below.
>
> > w1:
>
> For the first question, **our proposed CBC measure does not operate in isolation for handling noisy labels; instead, it functions in collaboration with the proposed TCL framework.** With the help of CBC, our TCL is a novel graph curriculum learning framework that can first learn clean easy nodes and then lean noisy hard nodes. This "easy-to-hard" training curriculum is effective on handling label noise and depend on the propose of CBC. Consequently, the proposed CBC, in collaboration with the TCL framework, effectively addresses label noise. The development of **both the TCL framework and the CBC measure are interdependent** and constitute the core contributions of our work.
>
> For the second question, we employed the CBC measure to identify class boundary nodes based on their topological structure. This measure **primarily utilizes topological structure, making it less susceptible to the impact of label noise**. This characteristic has been empirically demonstrated in our work, as evidenced in Figures 2 and 3 with different datasets, such as Cora and WikiCS. These figures, which can be viewed as toy experiments, clearly illustrate that **class boundary nodes are assigned high CBC scores even in scenarios with a high ratio of label noise**. Additionally, to further address concerns regarding this aspect of our research, we have included **an extra experiment in Figure 11 (Appendix E.2)**. In this experiment, we altered the structure of the dataset under label noise conditions, demonstrating that our CBC score can **consistently identify class boundary nodes across various datasets**.
>
>
> > w2:
>
> We appreciate the reviewer's question about this theory.
>
> In Theorem 1, we would like to kindly clarity that we used the widely acknowledged **local-dependence assumption** applicable to graph-structured data. This assumption aligns with Markov chain principles, stating that **the node is independent of the nodes that are not included in their two-hop neighbors when utilizing two-layer GNN, which does not means the totally i.i.d w.r.t. each node but means i.i.d w.r.t. subgroups**. The local-dependence assumption is well-established and has been **widely adopted in numerous graph theory studies** [1] [2] [3]. It endows models with desirable properties which make them amenable to statistical inference [2]. Consequently, this assumption forms a sound basis for our Theorem 1, which from risk-consistent statistical theory. The derived upper bound, based on this assumption, is both **reasonable and applicable** within the context of graph theory.
>
> To clarity the rationality behind the theory deduction, we have added more discussion with similar works for reference in the corresponding parts of the revised submission.
>
> [1] Wu, T., Ren, H., Li, P., & Leskovec, J. (2020). Graph information bottleneck. Advances in Neural Information Processing Systems, 33, 20437-20448.
>
> [2] Schweinberger, M., & Handcock, M. S. (2015). Local dependence in random graph models: characterization, properties and statistical inference. Journal of the Royal Statistical Society Series B: Statistical Methodology, 77(3), 647-676.
>
> [3] Didelez, V. (2008). Graphical models for marked point processes based on local independence. Journal of the Royal Statistical Society Series B: Statistical Methodology, 70(1), 245-264.

---

> ### Author Response · Authors · 2023-11-19
> **The response to Reviewer cJzR (Part 2)**
>
> > w3:
>
> The theorey of TCL is based on the binary classification setting and the postive/negative sample correspond to the nodes have postive/negative labels. **The error distirbution refects the difference between the noisy distirbution and the clean distirbution**. Essentially, this error distribution serves as a bridge connecting the noisy and clean distributions in our upper bound and we have **reinforced** this point in our theorem 1 of the revised submission.
>
> In the last two rows of our upper bound, the KL-divergence between the error distribution E and the clean distribution serves as an indicator of the divergence between the noisy and clean distributions. Essentially, the greater the deviation of the error distribution E from the clean distribution, the larger the difference between the noisy and clean distributions. This increased divergence directly increase the upper bound of the expected risk, making it more challenging to learn a proper classifier.
>
> Note that in the upper bound, the $0 < \alpha_{\lambda} <= 1$ is a decresing parameter, diminishing as the volume of training data increases in curriculum learning. Consequently, **training a classifier directly with the entire noisy dataset can lead to suboptimal generalization**. This is primarily due to the exacerbated deviation between the noisy and clean distributions. This reveals the significance of adopting an 'easy-to-hard' learning curriculum that first train model on a subset of easy clean node and progressively incorporating the noisy hard nodes. This training process effectively mitigates the adverse impact of the noisy distribution. Thus, this theory aligns with our proposed method, highlighting its relevance and effectiveness.
>
>
> >w4:
>
> Thank you very much for recommending these excellent works. Following the advice, we have added discussion about these three works in **the revised first paragraph of our introduction**. Furthermore, a detailed comparison between these works and our own research can be found in the part of **revised related work (Appendix B.3)**. We summarize the comparison as follows:
>
> The paper [1] and [2] are basd on the homophily assumption and employ the power of graph contrastive learning to learn robust node representations in the presence of label noise. The paper [3] employs the topological structure of the graph, along with a set of clean nodes, to mitigate label noise in graph data. These three approaches offers unique insights into handling label noise. **Different from these works,** we develop a novel graph curriculum learning framework that can learn easy clean nodes first and then learn from hard noisy nodes, with the aid of the proposed CBC measure that **robustly** identifies the learning difficulty of nodes in a graph with noisy labels. The design of the CBC measure is **novel** in the graph curriculum learning, marking it as a **significant contribution** in this field.
>
> Besides, as these works have not open their official codes, we are still reproducing them to complement the empirical comparison. Once we finished the experiments, we will place the results here and update in our submission. Thank you again for the recommendation.

---

> ### Author Response · Authors · 2023-11-21
>
> Dear reviewers cJzR,
>
> We have diligently addressed most of your concerns. If you still have concerns about our paper, please join the rolling discussion, where we await your valuable insights and comments.
>
> Thanks.

---

> ### Author Response · Authors · 2023-11-22
> **Looking forward to your reply**
>
> Dear Reviewer cJzR:
>
> Our sincere thanks go out to you for reviewing this paper! We do our best to address the issues raised by relevant work, experimental results and more empirical evidence. The discussion deadline is approaching now. Is there anything unclear in these explanations and descriptions?
>
> It would be greatly appreciated if your concerns had been addressed. However, if you require any further clarification, we can provide it before the deadline for discussion.
>
> Thanks!
>
> Authors

---

> > ### Comment · Reviewer_cJzR · 2023-11-23
> >
> > The author has partially addressed my concerns and I will keep my score unchanged

---

### Official Review · Reviewer_rtA1 · 2023-10-31

**Soundness:** 3 good
**Presentation:** 3 good
**Contribution:** 3 good
**Rating:** 5
**Confidence:** 3

**Summary:**

This paper proposes a metric called Class-conditional Betweenness Centrality, which is used for easy-to-hard sample selection in Curriculum Learning for graph data. Based on this, the authors design a Topological Curriculum Learning (TCL) framework and prove that it minimizes an upper bound of the expected risk.

**Strengths:**

1. The robustness and effectiveness of the Class-conditional Betweenness Centrality metric have been thoroughly and experimentally validated.

2. The CBC measure and the corresponding Topological Curriculum Learning are intuitive and meaningful in the graph curriculum learning.

**Weaknesses:**

1. The proposed Class-conditional Betweenness Centrality (CBC) is to some extent influenced by the homo. ratio of the dataset. Compared to Cora, conducting toy experiments with certain specifically synthesized data would be more accurate.

2. The paper lacks a more specific introduction to the impact of over-smoothing on Curriculum Learning for graph data, making it somewhat disjointed.

3. The improvements over existing CL methods are not significant.

**Questions:**

1. Fig.6 does not provide a comparison of effectiveness between other Curriculum Learning methods.
2. Is there any experimental results based on models that can effectively alleviate over-smoothing, used as the GNN backbone?
3. The features in Fig 2, are their features after being processed by GNN? If not, could authors provide that visualizations?
4. Will the proposed method be effective on homophily datasets.

---

> ### Author Response · Authors · 2023-11-19
> **The response to Reviewer rtA1 (Part 1)**
>
> We thank Reviewer rtA1 for reviewing our paper and providing helpful feedback on our work. We address many of Reviewer rtA1’s concerns in the general response above.
>
> > w1：
>
> Thank you for the suggestion. To answer the reviewer's question, we have included additional toy experiments in **Fig. 11 (Appendix E.2)** that vary homo. ratios. According to Fig. 11, a decrease in the homophily ratio results in an increased number of nodes near class boundaries, which consequently exhibit higher CBC scores. The experiments demonstrate that our CBC measure can effectively reflect the topology of nodes with different homophily ratios, even as the complexity of the graph increases, which reinforces its utility in diverse graph structures.
>
> > w2:
>
> Note that, this work take advantage of curriculum learning to handle the label noise on graph data. In Graph Neural Networks (GNNs), **"over-smoothing"** refers to the phenomenon where, as the network depth increases, node features become increasingly similar. This similarity **poses a challenge when employing curriculum learning with label noise**, making it difficult to distinguish between "easy" and "hard" nodes due to the homogenization of features caused by over-smoothing. Furthermore, this issue persists even in shallow GNNs, leading to under-confident predictions and complicating the establishment of an "easy-to-hard" training curriculum, as noted in previous studies [1][2]. Addressing this challenge, our work introduces a novel methd, which proposes **a robust CBC measure**. This measure effectively distinguishes between "easy" and "hard" nodes, taking into account the graph structure rather than the prediction of GNNs, thereby **mitigating the over-smoothing problem**.
>
> To address the concern, we have revised the paper by **emphasizing this aspect in the introduction** for readability.  Additionally, we have included **a more detailed discussion** regarding this part in the **related work (Appendix B.1)**, to provide a comprehensive understanding for the readers.
>
> [1] Wang, X., Liu, H., Shi, C., & Yang, C. (2021). Be confident! towards trustworthy graph neural networks via confidence calibration. Advances in Neural Information Processing Systems, 34, 23768-23779.
> [2] Hsu, H. H. H., Shen, Y., Tomani, C., & Cremers, D. (2022). What Makes Graph Neural Networks Miscalibrated?. Advances in Neural Information Processing Systems, 35, 13775-13786.
>
> >w3:
>
> The term "significant" indeed encompasses a broad assertion. First, it's crucial to highlight that in every cases, our approach **consistently outperforms** existing curriculum learning (CL) methods in terms of average accuracy. Second, **as the second-best baselines change in case-by-case noise levels, it is not reasonable to roughly consider the comparison between RCL with the case-by-case second-best baseline**. It is more fair to compare the **average improvement** between TCL and other baselines across all cases within a single dataset. These calculations, based on the data from our Tables 1 and 2, are **summarized in the table below**. The data presented in the table below illustrates the superiority of our method over the second-best approaches, with an significant improvement margin in all datasets. Notably, **while the second-best methods are different across various datasets, our method consistently emerges the top performance.** This evidence demonstrates that our approach substantially outperforms other baseline methods and is effective on noisy labeled graph.
>
> **The average improvement of the following table is calulated by averaging the accuracy improvement between TCL and other methods across all cases.** For example, as we can see, although Me-Momentum is competitive, TCL achieves 6.52\%, 9.91\%, and 18.29\% improvement on average on Facebook, Physics and DBLP datasets.
> |Baselines|  CORA   | CiteSeer  | Pubmed | WiKiCS | Facebook | Physics | DBLP |
> |----|  ----  | ----  | ----  | ----  | ----  | ----  |----  |
> |CP| 4.34  | 5.68 |1.49|4.51|**2.36**|**3.08**|7.72|
> |NRGNN| 3.11  |  **1.28**| 1.60|3.25|7.85|3.84|**1.55**|
> |PI-GNN| 2.40 |4.14|1.21|1.05|3.27|3.69|3.48|
> |Co-teaching+|4.18 |7.56|1.73|4.19|3.18|5.27|11.61|
> |Me-Momentum |**1.34** | 1.31|1.15|**0.97**|6.52|9.91|18.29|
> |MentorNet |5.11 | 8.98 |2.24|5.85|4.63|6.89|14.71|
> |CLNode  |6.22|10.73|**1.09** |1.38|2.91|3.42|3.74|
> |RCL  |15.83 |18.49|5.04|7.83|11.07|10.69|14.84|

---

> ### Author Response · Authors · 2023-11-19
> **The response to Reviewer rtA1 (Part 2)**
>
> > Q1:
>
> Thank you for your suggestion. Actually, the purpose of this section is to conduct an ablation study evaluating the "easy-to-hard" mechanism with or without the CBC measure (**TCL v.s. Vanilla**). Furthermore, **we follow the reviewer's advice and have included comparisons with two additional curriculum learning methods in the revision**. These experiments and results are now incorporated into **the Figure 10 in Appendix E.1**, which further highlights the advantages of our method.
>
>
> > Q2:
>
> We would like to kindly clarify some points. First, most models for alleviating over-smoothing are based on adjusting the connections between nodes. This adjustment heavily relies on the presence of correctly labeled nodes [1,2,3]. Therefore, these models are not suited in our setting scenarios. Addressing the over-smoothing issue in scenarios with label noise constitutes another research problem and does not impact the main contribution of our work.
>
> Besides, the primary contribution of our work is the development of a robust CBC measure. This measure leverages curriculum learning to effectively address label noise in graph data. Notably, the CBC measure is independent of trained models and relies solely on topological structure, which is unaffected by the 'over-smoothing' phenomenon and provides another way to distinguish the easy and hard nodes. Therefore, changing the GNN backbone in our experiments does not affect the evaluation of our main contribution.
>
> [1] Zhou, K., Huang, X., Li, Y., Zha, D., Chen, R., & Hu, X. (2020). Towards deeper graph neural networks with differentiable group normalization. Advances in neural information processing systems, 33, 4917-4928.
>
> [2] Chen, D., Lin, Y., Li, W., Li, P., Zhou, J., & Sun, X. (2020, April). Measuring and relieving the over-smoothing problem for graph neural networks from the topological view. In Proceedings of the AAAI conference on artificial intelligence (Vol. 34, No. 04, pp. 3438-3445).
>
> [3] Rusch, T. K., Chamberlain, B., Rowbottom, J., Mishra, S., & Bronstein, M. (2022, June). Graph-coupled oscillator networks. In International Conference on Machine Learning (pp. 18888-18909). PMLR.
>
> > Q3:
>
> Yes, as the understanding of the reviewer, the features in Fig. 2 are after being processed by GNN. We **have added the highlight of this point in Figure 2 caption in the revised submission**.
>
>
> > Q4:
>
> We would like to kindly point out that the most of experiment dataset are common-used homophily datasets, each exhibiting varying degrees of homophily. Across all cases, our method consistently outperforms baseline methods by achieving highest average accuracy.
>
> We guess that if the question of the reviewer is actually about whether our method could be effective on heterphily dataset. To address this concern, we have included three commonly-heterophilous datasets, e.g., DBLP [1], Chameleon [2], Squirrel [2] under 30% instance-dependent label noise. The summary of experiment results in the below table. Our method **still show the superior performance on heterphily datasets**.
>
> |dataset | DBLP   | Chameleon | Squirrel |
> |----|  ----  | ----  | ----  |
> |CP|70.02 | 55.08| 43.42|
> |NRGNN|72.48 |49.02 | 41.35|
> |PI-GNN|71.72 |52.85 |43.31 |
> |Co-teaching+| 66.32| 53.07 | 39.48 |
> |Me-Momentum |59.88  |55.01 | 44.38 |
> |MentorNet | 63.73 |53.73 |39.63 |
> |CLNode  | 72.32|52.85 |35.92 |
> |RCL  | 63.20|52.96 | 40.59 |
> |**TCL**| **74.70**|  **56.17** | **48.03** |
>
> [1] Fu, X., Zhang, J., Meng, Z., & King, I. (2020, April). Magnn: Metapath aggregated graph neural network for heterogeneous graph embedding. In Proceedings of The Web Conference 2020 (pp. 2331-2341).
>
> [2] Rozemberczki, B., Allen, C., & Sarkar, R. (2021). Multi-scale attributed node embedding. Journal of Complex Networks, 9(2), cnab014.

---

> > ### Author Response · Authors · 2023-11-22
> > **Looking forward to your reply**
> >
> > Dear Reviewer rtA1:
> >
> > Our sincere thanks go out to you for reviewing this paper! We do our best to address the issues raised by relevant work, experimental results and more empirical evidence. The discussion deadline is approaching now. Is there anything unclear in these explanations and descriptions?
> >
> > It would be greatly appreciated if your concerns had been addressed. However, if you require any further clarification, we can provide it before the deadline for discussion.
> >
> > Thanks!
> >
> > Authors

---

> ### Author Response · Authors · 2023-11-21
>
> Dear reviewers rtA1,
>
> We have diligently addressed most of your concerns. If you still have concerns about our paper, please join the rolling discussion, where we await your valuable insights and comments.
>
> Thanks.

---

### Author Response · Authors · 2023-11-19
**General Response by Authors**

We thank reviewers for their valuable feedback, and appreciate the great efforts made by all reviewers, ACs, SACs and PCs.

For readability, we will refer to the Reviewers rtA1, cJzR, GKC8
, and gbE5 as **R1, R2, R3, and R4**, respectively, **in the order displayed from top to bottom**.

We appreciate the positive evaluation from all reviewers, including: the problem is **well-motivated** (R4), the method is **novel**(R2,R3) and **reasonable** (R1,R3) , supported by **theoretical results** (R1,R2,R3) and **extensive experiments** (R1,R2,R3), **well-written**(R1,R2,R3,R4).

According to the advice, we have carefully revised our draft with the proofreading to correct some typos and mistakes. In the following, we provide a summary of our updates, and for detailed responses, please refer to the feedback of each comment/question point-by-point.


- We have incorporated a comprehensive **discussion and comparison** involving some related domains and methods, including the over-smoothing problem in our setting (R1), the generlization of our method on heterogeneous graphs(R1, R3), discussion about the latest related works (R2), the generlization of our method on OOD problem (R3), more related work on graph curriculum learning (R4).
- We have provided **clarification and revisions** to address potential misunderstandings and confusion, including illustration to the impact of over-smoothing (R1),  the connection between CBC and label noise (R2), the correlation between CBC and class boundary nodes (R2), an explanation of Theorem 1 (R2), explicit statements of assumptions (R2), advancement of the proof (R3), clarification of the pretrained model (R3), further interpretation of experimental result (R1,R4), the importance of CBC on TCL framework (R4).
- We have **enriched our experiments** based on the existing extensive results, including the underlying mechanism of TCL (R1, Fig.10), the CBC with vary homophily ratio (R1, R2, R3, Fig.11), TCL performance comparsion on heterphily datasets (R1, R3, Table 7). (Appendix E.1-E.3)


The above updates in the revised version (the regular part and the appendix of totally **30 pages**) are highlighted in red color.

We appreciate all reviewers’ time again. We are looking forward to your reply!

---

### Meta-Review · Area_Chair_g9TW · 2023-12-04

**Metareview:**

I have read all the materials of this paper including the manuscript, appendix, comments, and response. Based on collected information from all reviewers and my personal judgment, I can make the recommendation on this paper, *reject*. No objection from reviewers who participated in the internal discussion was raised against the reject recommendation.

**Research Question**

The authors consider the curriculum learning for node classification.

**Motivation & Challenge Analysis**

The authors pointed two challenges of conducting curriculum learning on GNNs. (1) The inherent over-smoothing effect in GNNs usually induces the under-confident prediction, which exacerbates the discrimination difficulty between easy and hard samples; (2) There is no available measure that considers the graph characteristic to promote informative sample selection in curriculum learning. The words "easy," "hard," and "informative" are too general to me. It would be better to provide some formal or informal definitions. Based on my understanding, the easy/hard samples are algorithm-dependent.

**Philosophy**

The authors aim to incorporate the topological information for easy-or-hard sample selection. This makes sense to me, but when comparing with the above challenges, we can see that there is no word like graph topology mentioned in the targeted challenges. I suggest the authors modify the descriptions and make the paper more coherent.

**Techniques**

The authors proposed a Class-conditional Betweenness Centrality (CBC) measure to efficiently distinguish the easy and hard nodes with topological enhancement for curriculum learning, which is based on personalized PageRank.

Figure 2 and 3 are used to verified the effectiveness of CBC when facing noisy labels. Where are decision boundaries? How to define how far or near a sample is to the decision boundaries? Figures 2-4 can only demonstrate the nodes with high CBC are far away from the decision boundary, why they can help on the curriculum learning is unclear. The authors held the assumption that "the CBC measure drives the TCL to select clean and easy nodes located far from class boundaries." This assumption needs to be verified in the experimental part.

Based on CBC, the authors proposed TCL for node classification, which follows the standard procedure of curriculum learning.

**Theoretical Analysis**

The theoretical analysis is general for curriculum learning on GNN, which is not particularly designed for TCL.

**Experiments**

1. Compared with the results in Table 1&2, Table 4 is the key results to demonstrate the effectiveness of the proposed measure, which compares different sample selection strategies.

2. In general, the experimental results are extensive and significant. I suggest the authors add statistical tests.

**Summary**

The proposed CBC is mainly based on PageRank or some embedding techniques, which is straightforward to incorporate topological information into consideration. Moreover, the challenges and philosophy do not match well.

**Justification For Why Not Higher Score:**

Although this paper seems self-standing, more efforts in terms of logic, novelty, and evaluation are needed to reach the bar of ICLR.

**Justification For Why Not Lower Score:**

N/A

---

### Decision · Program_Chairs · 2024-01-16

Reject